# CCR5 is a suppressor for cortical plasticity and hippocampal learning and memory

**Miou Zhou[1†], Stuart Greenhill[2,3†], Shan Huang[1], Tawnie K Silva[1], Yoshitake Sano[1,4], Shumin Wu[5], Ying Cai[1], Yoshiko Nagaoka[5], Megha Sehgal[1], Denise J Cai[1], Yong-Seok Lee[1,6], Kevin Fox[2*], Alcino J Silva[1*]**

[1]Departments of Neurobiology, Psychology, Psychiatry, Integrative Center for Learning and Memory and Brain Research Institute, University of California, Los Angeles, Los Angeles, United States; [2]Cardiff School of Biosciences, Cardiff University, Cardiff, United Kingdom; [3]School of Life and Health Sciences, Aston University, Birmingham, United Kingdom; [4]Department of Applied Biological Science, Tokyo University of Science, Chiba, Japan; [5]Department of Molecular and Medical Pharmacology, University of California, Los Angeles, Los Angeles, United States; [6]Department of Physiology, Seoul National University College of Medicine, Seoul, Republic of Korea

**\*For correspondence:** foxkd@ cardiff.ac.uk (KF); silvaa@mednet. ucla.edu (AJS)

[†]These authors contributed equally to this work

**Competing interests:** The authors declare that no competing interests exist.

**Abstract** Although the role of CCR5 in immunity and in HIV infection has been studied widely, its role in neuronal plasticity, learning and memory is not understood. Here, we report that decreasing the function of CCR5 increases MAPK/CREB signaling, long-term potentiation (LTP), and hippocampus-dependent memory in mice, while neuronal CCR5 overexpression caused memory deficits. Decreasing CCR5 function in mouse barrel cortex also resulted in enhanced spike timing dependent plasticity and consequently, dramatically accelerated experience-dependent plasticity. These results suggest that CCR5 is a powerful suppressor for plasticity and memory, and CCR5 over-activation by viral proteins may contribute to HIV-associated cognitive deficits. Consistent with this hypothesis, the HIV V3 peptide caused LTP, signaling and memory deficits that were prevented by Ccr5 knockout or knockdown. Overall, our results demonstrate that CCR5 plays an important role in neuroplasticity, learning and memory, and indicate that CCR5 has a role in the cognitive deficits caused by HIV.

## Introduction

C-C chemokine receptor 5 (CCR5) is a seven-transmembrane G protein-coupled receptor (GPCR) involved in recruiting leukocytes to sites of tissue damage during inflammatory responses. CCR5 is highly expressed in T cells and macrophages in the immune system (*Sorce et al., 2011*). In the central nervous system, CCR5 is expressed in microglia, astrocytes and neurons in multiple brain regions (*Cartier et al., 2005*; *Meucci et al., 1998*; *Tran et al., 2007*; *Westmoreland et al., 2002*), including the CA1 region of the hippocampus (*Torres-Muñoz et al., 2004*).

Ligand binding to CCR5 is known to modulate several parallel signaling cascades implicated in learning and memory, including the suppression of adenylyl cyclase (AC), as well as the activation of the PI3K/AKT and p44/42 MAPK signaling (*Cartier et al., 2005*; *Paruch et al., 2007*; *Tyner et al., 2005*). The CCR5 endogenous ligand RANTES (also known as CCL5) is reported to block neuronal $[Ca^{2+}]_i$ oscillations (*Meucci et al., 1998*) and to modulate glutamate release (*Musante et al., 2008*), suggesting a role for CCR5 in the regulation of neuronal function.

CCR5 has a key role in HIV infection by mediating virus cellular entry. *Ccr5* knockdown inhibits HIV-1 infection in macrophages (*Liang et al., 2010*) and CCR5 antagonists effectively reduce HIV-1

expression in AIDS patients (*Hunt and Romanelli, 2009*). Treatment with the CCR5 antagonist maraviroc has been reported to improve neurocognitive test performance among patients with moderate cognitive impairment (*Ndhlovu et al., 2014*), supposedly by reducing monocytes and inflammation. Cognitive deficits affect approximately 30% of HIV-positive adults and 50% of HIV-positive infants (*Galicia et al., 2002*), and are a significant clinical problem associated with HIV infection. Although CCR5 plays a crucial role in HIV infection in the central nervous system (CNS) (*Ellis et al., 2007*; *Zhou and Saksena, 2013*), little is known about its role in neuronal plasticity or learning and memory, or whether this receptor plays a direct role in HIV-associated cognitive disorders.

Here, we demonstrate that manipulations that decrease CCR5 function result in elevated MAPK and CREB levels during learning, enhance synaptic plasticity and improve both cortical sensory plasticity and hippocampal learning and memory, while the transgenic overexpression of this receptor causes learning and memory deficits. Thus, our studies reveal an important suppressor role for CCR5 in neuroplasticity, learning and memory, independent of the proposed roles of CCR5 in neuroinflammation and neurodegeneration. Since our results and other studies (*Cormier and Dragic, 2002*; *Morikis et al., 2007*; *Shen et al., 2000*) show that HIV coat proteins can bind and activate CCR5, thus activating its memory suppressor functions, our results suggest that besides the neuroinflammation induced neurodegeneration that can cause HIV-associated cognitive deficits, CCR5 activation by HIV coat proteins also contributes to the cognitive deficits caused by HIV.

## Results

### Identification of *Ccr5* knockout mice in a reverse genetic memory screen

Our first indication that CCR5 was involved in plasticity and memory came from a reverse genetic memory screen. In total, 148 transgenic and knockout mutant mouse strains with controlled genetic backgrounds were chosen at random from the inventories of commercial vendors (Jackson Laboratories and Taconic Farms) as well as individual laboratories, and screened for contextual memory phenotypes (*Figure 1—figure supplement 1* and *Figure 1—source data 1*). Interestingly, 6 out of the 8 chemokine or chemokine receptor mutant strains screened (# labeling, *Figure 1—figure supplement 1*) showed positive Z scores (see Materials and methods), including three mutant strains with scores above 1, suggesting enhanced memory for contextual conditioning. HRas$^{G12V}$ mice were used for comparison, since they were previously reported (*Kushner et al., 2005*) to show enhanced memory for contextual fear conditioning. Like HRas$^{G12V}$ mice, mice homozygous for a null-mutation of the *Ccr5* gene (*Ccr5$^{-/-}$*) showed average Z scores above 1. The *CCR5* knockout resulted in memory enhancements for contextual conditioning when tested 24 hr after training (*Figure 1—figure supplement 2A*, $t_{(14)}$ = 2.43 p<0.05, Student's *t*-test) and 2-weeks after training (*Figure 1—figure supplement 2B*, $t_{(14)}$ = 3.07 p<0.01, Student's *t*-test). These results show that the *Ccr5* knockout results in enhanced long-term (24 hr) and remote (2-weeks) memory. Importantly, activity levels including baseline activity and activity bursts during shock-exposure were normal (*Figure 1—figure supplement 2C,D*), a result demonstrating that the enhanced freezing of *Ccr5$^{-/-}$* mice is not due to either decreases in activity or increased sensitivity to the unconditioning stimulus (i.e., the footshock). In contrast to contextual conditioning, tone conditioning (*Hall et al., 2001*) was unaltered in the *Ccr5$^{-/-}$* mice (*Figure 1—figure supplement 2E*), confirming that the contextual conditioning enhancement of these mutants is not due to non-specific behavioral changes that alter conditioning responses.

### *Ccr5$^{+/-}$* mice show enhanced memory in multiple memory tasks

To test whether the heterozygous deletion mutation (*Ccr5$^{+/-}$*) also results in enhanced memory, *Ccr5$^{+/-}$* mice were trained in the contextual fear conditioning task and were tested 2-weeks after training. Compared to WT littermates, *Ccr5$^{+/-}$* mice showed increased freezing levels (*Figure 1A*, $t_{(20)}$ = 2.37 p<0.05, Student's *t*-test); in contrast, baseline activity and shock sensitivity were normal in these mutants (*Figure 1—figure supplement 3A*), demonstrating that both the *Ccr5* heterozygous and homozygous mutations result in enhanced contextual memory.

To determine whether the *Ccr5$^{+/-}$* mutation also affects other forms of hippocampal-dependent memory (*Kogan et al., 2000*; *Riedel et al., 1999*), the mice were tested in the Morris water maze and social recognition tasks. In the hidden-platform version of the Morris water maze, mice were

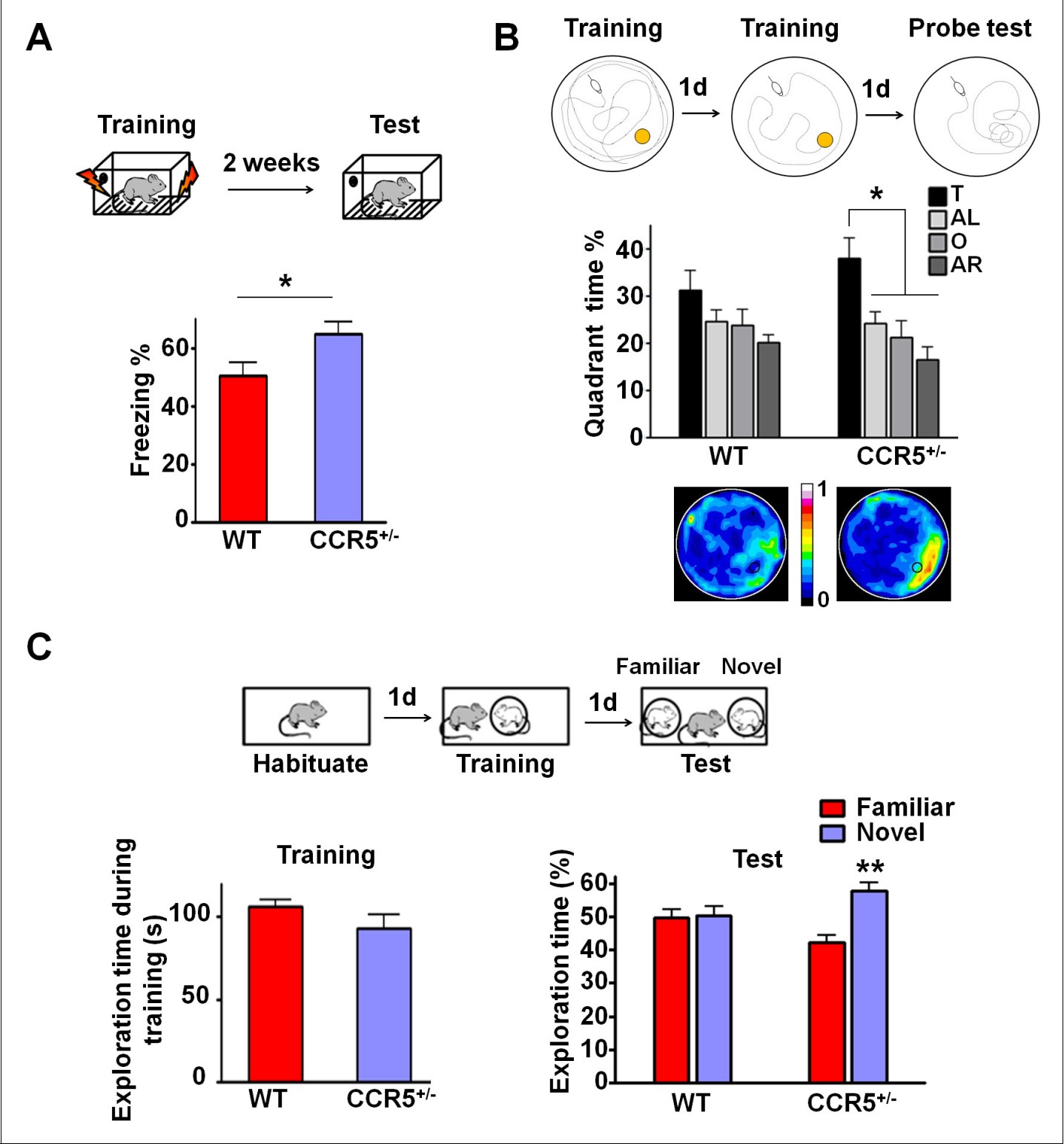

**Figure 1.** *Ccr5*<sup>+/-</sup> mice show enhanced memory in multiple memory tasks. (**A**) In a fear conditioning test given two weeks after training, *Ccr5*<sup>+/-</sup> mice showed enhanced contextual memory (WT n = 11, *Ccr5*<sup>+/-</sup> n = 11; *p<0.05, Student's *t*-test). (**B**) In a water maze probe test given after two days of training, *Ccr5*<sup>+/-</sup> mice spent significantly more time in the target quadrant than in the other three quadrants of the water maze. In contrast, WT mice did not search selectively for the platform (WT n = 14, *Ccr5*<sup>+/-</sup> n = 16; *p<0.05, Two-way ANOVA with repeated measure). Heat maps below the bar graphs show the combined traces of the mice from each group during the probe test. (**C**) There was no difference between WT mice and *Ccr5*<sup>+/-</sup> mice in social

*Figure 1 continued on next page*

*Figure 1 continued*

training. In the social recognition test given 24 hr after training (7 min, a training time chosen to undertrain WT mice), $Ccr5^{+/-}$ mice, but not WT mice, spent more time exploring the novel OVX mouse (WT n = 14, $Ccr5^{+/-}$ n = 19; **p<0.01, one sample *t*-test compared to 50%). Error bars indicate SEM.

The following source data and figure supplements are available for figure 1:

**Source data 1.** Full names and MGI accession # of the 148 mutant strains in the reverse genetic memory screen (*Jax stock#: 370200; **Jax stock#: 370202).

**Figure supplement 1.** Identification of *Ccr5* knockout mice in a reverse genetic memory screen.

**Figure supplement 2.** $Ccr5^{-/-}$ mice show enhanced contextual memory.

**Figure supplement 3.** Behavioral tasks to test the baseline activity and activity burst during fear conditioning, and anxiety and locomotive levels in WT and $Ccr5^{+/-}$ mice.

**Figure supplement 4.** WT and $Ccr5^{+/-}$ mice show similar GFAP and TUNEL immunostaining and similar spine density.

tested for their ability to use spatial cues around a pool to find an escape platform hidden just beneath the water surface. Following 2-days of training, memory was assessed by a probe test wherein the mice search for 60 s with the platform removed from the pool. Since this probe test was administered early in training, the WT mice did not search selectively for the platform. In contrast, even with this limited amount of training, $Ccr5^{+/-}$ mice spent significantly more time in the quadrant where the platform had been during training (target quadrant) than the other three quadrants, demonstrating that the $Ccr5^{+/-}$ mutation not only enhances learning and memory for contextual conditioning, but also results in enhanced spatial learning and memory [*Figure 1B*; Two-way ANOVA with repeated measure, overall (genotype × percentage of time in each quadrant) interaction: $F_{(3,84)}$ = 0.81; Main effect of quadrant%: $F_{(3,84)}$ = 6.99. Between the target quadrant and all other quadrants for $Ccr5^{+/-}$ mice: p<0.05, Bonferroni post-tests]. Heat maps, derived from the combined swimming traces of the mice in each group during the probe test, also illustrate the spatial learning and memory enhancement of $Ccr5^{+/-}$ mice (*Figure 1B*).

The social recognition task takes advantage of mice's preference for novel stimuli and tests their ability to distinguish familiar versus novel conspecifics (*Kogan et al., 2000*). For this task, mice were first habituated to the testing chamber on day 1; on day 2, they were placed in the same chamber and allowed to interact for 7 min with an ovariectomized (OVX) female mouse placed under a wired cylinder (training session). On day 3, individual mice were placed back into the same chamber (test session), and allowed to interact with two OVX females (one familiar and one novel) (*Figure 1C*). In the test session, compared to WT mice, $Ccr5^{+/-}$ mice spent significantly more time exploring the novel OVX mouse, indicating that the $Ccr5^{+/-}$ mutation also enhanced learning and memory in the social recognition test (*Figure 1C*, $t_{(18)}$ = 3.39 p<0.01, one sample paired *t*-test compared to 50%). Importantly, the $Ccr5^{+/-}$ mutation did not affect the total interaction time during training, demonstrating that their learning and memory enhancement is not due to increased social interaction.

Since anxiety and activity levels could confound the results of the memory tests described above, we tested the $Ccr5^{+/-}$ mice and their WT littermate controls in the elevated plus maze and open field tasks. $Ccr5^{+/-}$ mice were similar to WT littermates in both open arm entries and percentage time spent in the open arms of the elevated plus-maze, two well-known measures of anxiety (*Figure 1— figure supplement 3B*); Analyses of the open field (e.g., measures of total distance traveled and percentage time in center zone) also revealed no differences between the $Ccr5^{+/-}$ mice and their WT controls (*Figure 1—figure supplement 3C*). A previous study reported that CCR5 deficiency results in the activation of astrocytes, which leads to neurodegeneration in aged mice (*Lee et al., 2009*). We measured astrocyte numbers and apoptosis with GFAP and TUNEL staining in the hippocampus of 3-month old mice, and found that in our genetic background there is no difference between WT and $Ccr5^{+/-}$ mice (*Figure 1—figure supplement 4A–C*). Measurements of spine density in the hippocampal CA1 subregion revealed no differences between YFP/WT and YFP/$Ccr5^{+/-}$ mice (*Figure 1— figure supplement 4D*). Altogether, these results indicate that the learning and memory

enhancements of the $Ccr5^{+/-}$ mice are not confounded by abnormal anxiety, changes in locomotor activity, astrogliosis, apoptosis, or spine density.

## Enhanced MAPK and CREB signaling and long-term potentiation in $Ccr5^{+/-}$ mice

Mitogen-activated protein kinases p44/42 (MAPKs) (*Atkins et al., 1998*; *Kushner et al., 2005*; *Schafe et al., 2000*) and cAMP-responsive element-binding protein (CREB) (*Bourtchuladze et al., 1994*; *Dash et al., 1990*; *Yin et al., 1994*) are known to have a central role in hippocampal learning and memory and in cortical plasticity (*Barth et al., 2000*; *Glazewski et al., 1999*). We focused our signaling studies at two different time points (1 hr and 3 hr) following training (*Chwang et al., 2006*; *Stanciu et al., 2001*). Prior to learning (measurements in home cage controls), there were no differences between WT and $Ccr5^{+/-}$ mice in either hippocampal MAPK or CREB activation measured with phospho-specific antibodies (*Figure 2A*). In contrast, $Ccr5^{+/-}$ mice showed enhanced phosphorylated MAPK (*Figure 2B*, $t_{(10)}$ = 4.45 p<0.01, Student's *t*-test) and enhanced phosphorylated CREB levels (*Figure 2C*, $t_{(12)}$ = 2.21 p<0.05, Student's *t*-test) at 1 and 3 hr after fear conditioning, respectively (see *Figure 2—figure supplements 1*, *2* and *3* for MAPK and CREB levels at both 1 hr and 3 hr).

Since MAPK/CREB signaling in neurons is known to affect synaptic plasticity, we tested whether the $Ccr5^{+/-}$ mice show enhanced long-term potentiation (LTP), a cellular mechanism underlying learning and memory (*Bliss and Collingridge, 1993*; *Lee and Silva, 2009*). Field EPSPs (fEPSPs) evoked by Schaffer collateral stimulation were recorded in the CA1 region of acute hippocampal slices prepared from WT and $Ccr5^{+/-}$ mice. Analyses of the fEPSPs between 50 and 60 min post-tetanus revealed enhanced LTP in $Ccr5^{+/-}$ mice (*Figure 2D*, $t_{(12)}$ = 2.61 p<0.05, Student's *t*-test). Our results suggest that the enhancement in MAPK/CREB signaling causes the hippocampal enhancements in LTP that likely underlies the hippocampal-dependent learning and memory enhancements of $Ccr5^{+/-}$ mice. More importantly, these signaling, electrophysiological and behavioral results of $Ccr5^{+/-}$ mice suggest that CCR5 is a plasticity and memory suppressor (*Abel et al., 1998*).

## Knockdown of *Ccr5* in adult hippocampus results in enhanced memory

The *Ccr5* knockout we studied is neither restricted to the hippocampus nor specific to adult brain neurons, leaving open a number of other alternative explanations for the results described above. Therefore, we used Adeno-Associated Viral vectors (AAV5) to restrict a shRNA-mediated *Ccr5* knockdown to the adult hippocampus. AAV5 vectors containing either shRNA-CCR5 or shRNA-dsRed (shRNA-Cont) (*Figure 3A*) were injected into the hippocampal pyramidal fields of 3 month-old C57BL/6N mice. One month after shRNA-CCR5 virus injection, *Ccr5* mRNA was markedly reduced in the hippocampus compared to shRNA-Cont virus (*Figure 3B*, $t_{(7)}$ = 7.54 p<0.001, Student's *t*-test). Similarly, in cultured HEK293 cells, infection of shRNA-CCR5, but not shRNA-Cont reduced *Ccr5* expression (*Figure 3—figure supplement 1A*). Additionally, one-month after shRNA-Cont or shRNA-CCR5 AAV injection, GFP expression (indicating AAV transfection) was observed widely in the hippocampal pyramidal fields (including CA1 and CA2) of both groups (*Figure 3C*). Most GFP-positive cells were recognized by the NeuN antibody (neuronal marker), but not by the Iba1 antibody (microglia marker) (*Figure 3D* and *Figure 3—figure supplement 1B*) or the GFAP antibody (astrocyte marker) (*Figure 3—figure supplement 1C*), demonstrating that the viral vectors we used mainly transfected neurons, a result that we also confirmed in the barrel cortex (see results below).

To test the impact of *Ccr5* knockdown in hippocampal pyramidal fields, we trained the transfected mice with contextual fear conditioning one-month after virus injection. When tested 2-weeks after training, the shRNA-CCR5 mice showed enhanced contextual memory compared to shRNA-Cont mice (*Figure 3E*, $t_{(15)}$ = 3.46 p<0.01, Student's *t*-test). Mice transfected with shRNA-CCR5 showed similar enhancements in spatial learning and memory in the Morris water maze. Specifically, we utilized the paradigm used for our knockout studies where a probe test was given after 2 days training, when most control mice (transfected with shRNA-Cont virus) still failed to learn the task (e. g., spent similar times in all four quadrants of the maze). Strikingly, mice transfected with the shRNA-CCR5 virus spent significantly more time in the target quadrant than the other three quadrants [*Figure 3F*; Two-way ANOVA with repeated measure, overall (virus × percentage of time in each quadrant) interaction: $F_{(3,120)}$ = 4.18; Main effect of quadrant%: $F_{(3,120)}$ = 9.45. Between the

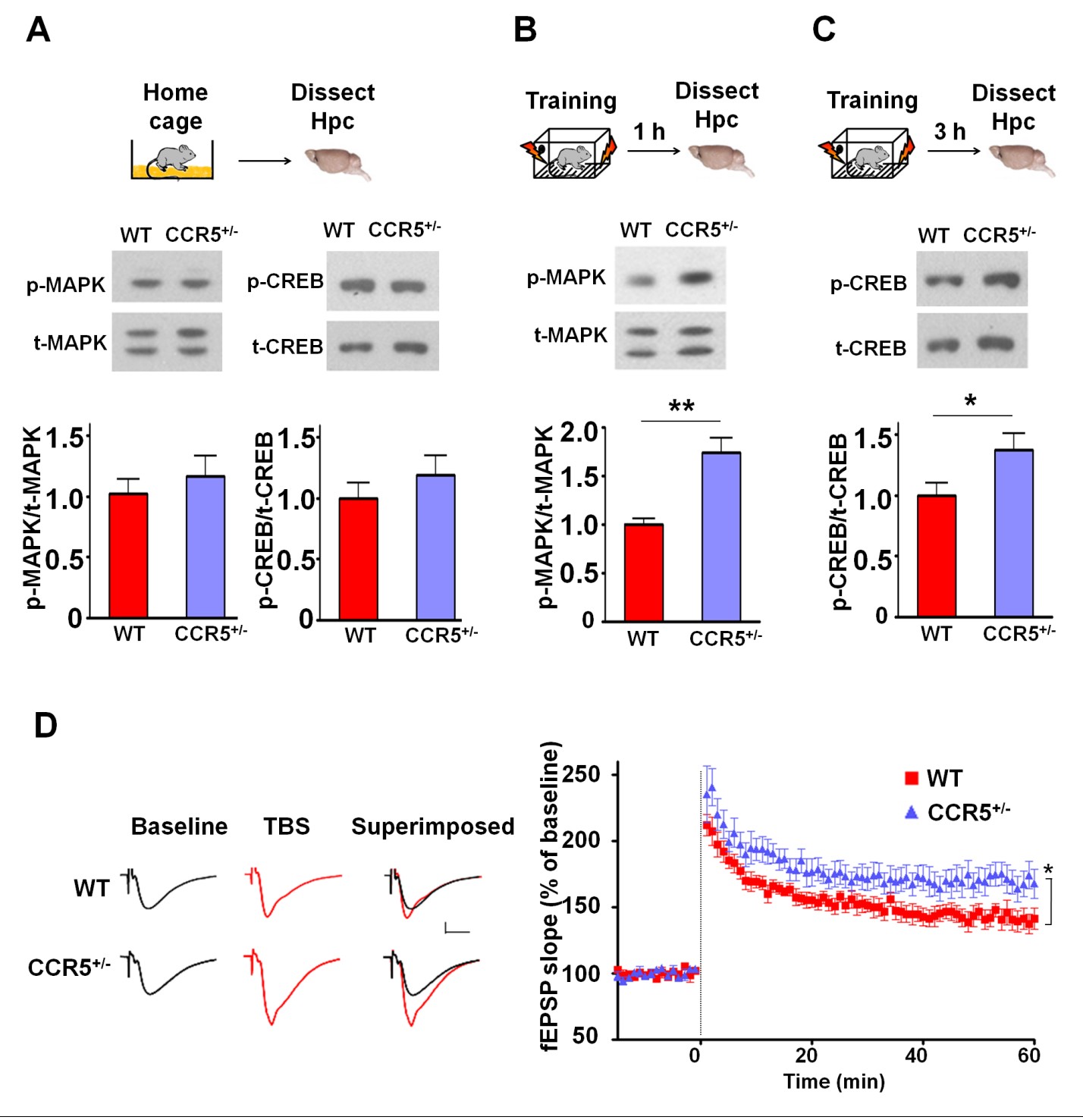

**Figure 2.** *Ccr5+/-* mice show post-training increases in MAPK and CREB signaling and enhanced hippocampal LTP. (A) Hippocampus collected from home cage WT and *Ccr5+/-* mice had similar levels of phosphorylated p44/42 MAPK and CREB (MAPK: WT n = 4, *Ccr5+/-* n = 4; CREB: WT n = 6, *Ccr5+/-* n = 7). (B) *Ccr5+/-* mice showed enhanced levels of phosphorylated p44/42 MAPK (normalized with total MAPK) in hippocampal samples collected one hour after fear conditioning training (WT n = 6, *Ccr5+/-* n = 6; **p<0.01, Student's *t*-test). (C) *Ccr5+/-* mice showed enhanced levels of phosphorylated CREB (normalized with total CREB) in hippocampal samples collected 3 hr after fear conditioning training (WT n = 7, *Ccr5+/-* n = 7; *p<0.05, Student's *t*-test). (D) CA1 fEPSPs were recorded in hippocampal slices before (baseline) and after 5 TBS (theta bursts stimulation, each burst consists of four stimuli at 100 Hz, 200 ms inter-burst interval). *Ccr5+/-* slices show a significant LTP enhancement in the fEPSP measured during the last 10 min of recordings following the tetanus (n = 7 mice for both WT and *Ccr5+/-*; *p<0.05, Student's *t*-test). Error bars indicate SEM.

*Figure 2 continued on next page*

*Figure 2 continued*

The following figure supplements are available for figure 2:

**Figure supplement 1.** Hippocampal MAPK/CREB signaling of WT, *Ccr5*^+/- and *Ccr5*^-/- mice.

**Figure supplement 2.** Hippocampal MAPK/CREB signaling of WT, *Ccr5*^+/- and *Ccr5*^-/- mice after fear conditioning.

**Figure supplement 3.** Hippocampal MAPK/CREB signaling of WT, *Ccr5*^+/- and *Ccr5*^-/- mice after fear conditioning.

target quadrant and all other quadrants for shRNA-CCR5 mice: p<0.001, Bonferroni post-tests; Target quadrant percentage between shRNA-Cont and shRNA-CCR5 group: $t_{(40)}$ = 2.53 p<0.05, Student's *t*-test]. Similar results were also obtained for other measures of probe trial performance including proximity to the target platform and platform crossings (data not shown). Heat maps, derived from the combined swimming traces of the mice in each group during the probe test, also illustrate the spatial learning and memory enhancement of the shRNA-CCR5 group (*Figure 3F*). To test whether with more extended training shRNA-Cont mice could also learn the water maze task, a second probe test was given after 5 days of training. In this probe test, both shRNA-Cont and shRNA-CCR5 mice spent more time in the target quadrant (*Figure 3—figure supplement 1D*), demonstrating that with additional training the shRNA-Cont mice are able to show spatial learning in the water maze task.

Altogether, these results demonstrate that CCR5 is a memory suppressor and that knocking-down *Ccr5* specifically in the adult CA1/CA2 is sufficient to enhance hippocampus-dependent learning and memory.

## CCR5 overexpression leads to learning and memory deficits

Our CCR5 knockout and knockdown results demonstrate that CCR5 is a plasticity and memory suppressor, and therefore predict that increases in CCR5 function lead to memory deficits. To test this hypothesis, we generated transgenic (Tg) mice overexpressing *Ccr5* under the *Camk2a* promoter (*Mayford et al., 1996*). Compared to WT mice, CCR5 Tg mice show increases in hippocampal *Ccr5* mRNA expression (*Figure 4A*). To test the impact of CCR5 overexpression on learning and memory, WT and CCR5 Tg mice were tested in three hippocampal-dependent learning tasks, including two contextual fear conditioning paradigms that are sensitive to changes in hippocampal function (*Cui et al., 2008*; *Matus-Amat et al., 2004*). Analyses of a context pre-exposure fear-conditioning paradigm (*Matus-Amat et al., 2004*) demonstrated that CCR5 Tg mice showed robust memory deficits (*Figure 4B*, $t_{(26)}$ = 3.71 p=0.001, Student's *t*-test). Similar results were also obtained with a multi-day fear conditioning training procedure (*Figure 4—figure supplement 1A*).

WT and CCR5 Tg mice were also trained in the Morris water maze, and probe tests were given after 3 days of training. In the probe test only WT mice, but not CCR5 Tg mice spent more time in the target quadrant than the other three quadrants [*Figure 4C*, Two-way ANOVA with repeated measure, Overall (genotype × quadrant%) interaction: $F_{(3,102)}$ = 3.42; Main effect of quadrant%: $F_{(3,102)}$ = 21.44; Between the target quadrant and all other quadrants for WT mice: ***p<0.001, Bonferroni post-tests; Target quadrant percentage between WT and *Ccr5* transgenic group: $t_{(34)}$ = 2.33 *p<0.05, Student's *t*-test]. CCR5 Tg mice also had higher average proximity from the platform and less target platform crossing compared to WT mice (*Figure 4—figure supplement 1B*), indicating that CCR5 overexpression results in spatial memory deficits. When a second probe test was given after extended training, both WT and CCR5 Tg mice spent more time in the target quadrant than the other three quadrants (*Figure 4—figure supplement 1C*), demonstrating that with extended training CCR5 Tg mice also learned the water maze task. Altogether, the results presented demonstrate a critical role for CCR5 in learning and memory.

We tested whether CCR5 overexpression changes either MAPK or CREB signaling in CCR5 Tg mice after fear conditioning. Compared to their WT controls, CCR5 Tg mice show similar phosphorylated p44/42 MAPK levels at baseline or after fear conditioning training (*Figure 4—figure supplement 2A, B and C*). Although CCR5 Tg mice and their WT controls have similar phosphorylated

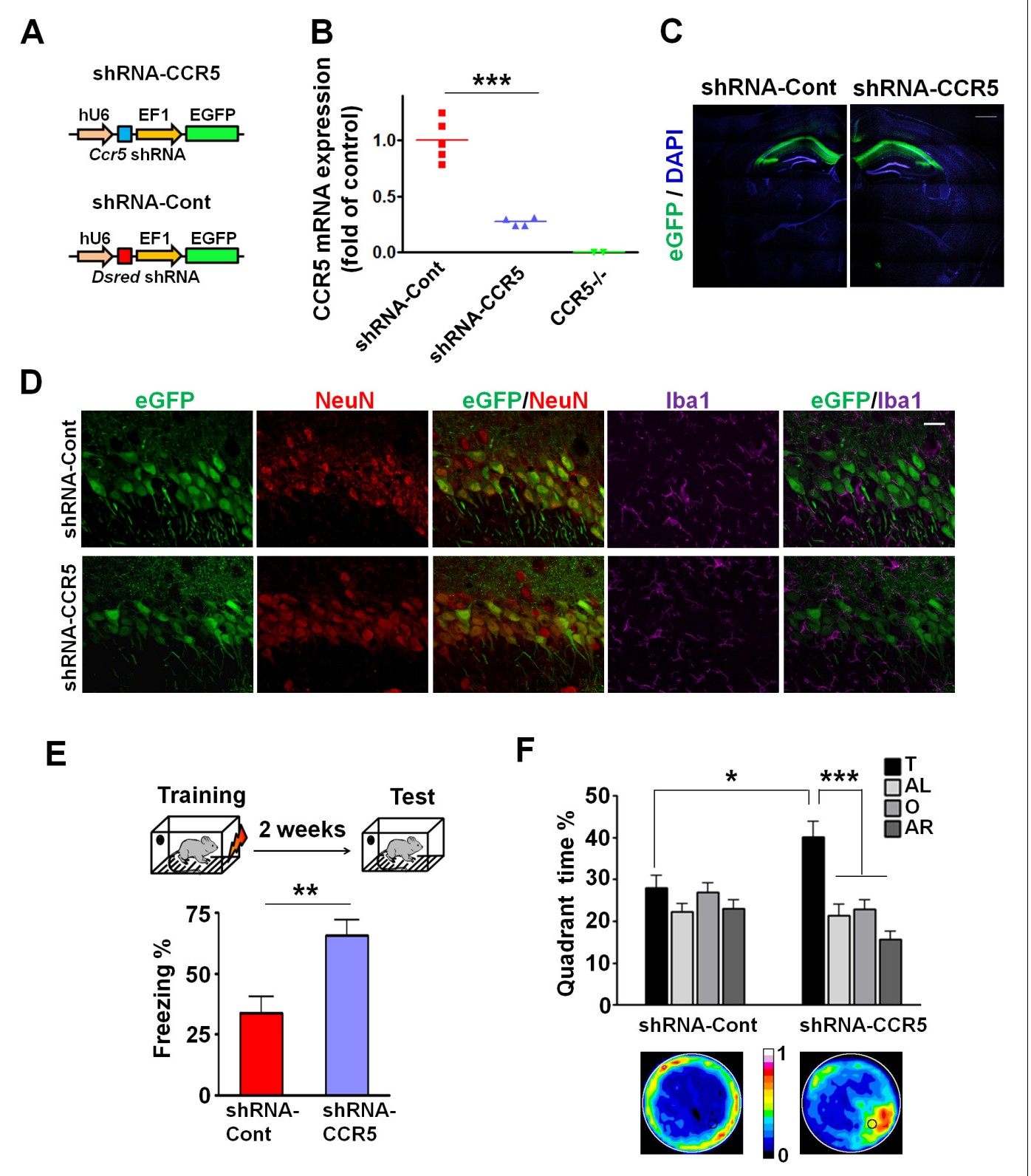

**Figure 3.** *Ccr5* knockdown in the adult hippocampus results in enhanced memory. (**A**) Schematics of the shRNA-CCR5 and shRNA-dsRed (shRNA-Cont) plasmids. (**B**) AAV containing shRNA-CCR5 or shRNA-Cont was injected into the hippocampus CA1/CA2 region, and dorsal CA1/CA2 *Ccr5* mRNA was measured one-month after virus injection. Compared to shRNA-Cont, shRNA-CCR5 triggered a significant reduction in *Ccr5* mRNA expression (***p<0.001, student's *t*-test). Also, *Ccr5*[-/-] mice showed no *Ccr5* mRNA expression in hippocampus. (**C**) AAV containing shRNA-CCR5 or shRNA-Cont

*Figure 3 continued on next page*

*Figure 3 continued*

was injected into the hippocampus. One month after virus injection, brain slices were stained with DAPI (nuclear labeling) and GFP (virus infection). Scale bar, 500 μm. (D) AAV containing shRNA-CCR5 or shRNA-Cont was injected into the hippocampus. One month after virus injection, brain slices were stained with GFP (virus infection), NeuN (neurons), and Iba1 (microglia). GFP was exclusively expressed in neurons. Scale bar, 20 μm. (E) AAV containing shRNA-CCR5 or shRNA-Cont was injected into the hippocampus, and mice were subjected to behavioral testing one month after virus injection. In the fear conditioning test, shRNA-CCR5 mice showed enhanced contextual memory when compared to those injected with shRNA-Cont virus (shRNA-cont n = 7, shRNA-CCR5 n = 10; **p<0.01, Student's *t*-test). (F) In the probe test given after two days of water maze training, only shRNA-CCR5 mice but not shRNA-Cont mice spent significantly more time in the target quadrant than the other three quadrants (shRNA-cont n = 22; shRNA-CCR5 n = 20; *p<0.05, Student's *t*-test; ***p<0.001, Two-way ANOVA with repeated measure). Heat maps below the bar graphs show the combined traces of the mice from each group during the probe test. shRNA-Cont mice showed a pattern of wall-hugging swim after two days training, but they learned the water maze task with extended training (*Figure 3—figure supplement 1D*). Error bars indicate SEM.

The following figure supplement is available for figure 3:

**Figure supplement 1.** shRNA-CCR5 knockdown efficiency in HEK 293 cells, AAV infection specificity, and mice performance in the water maze probe test 2.

CREB levels at baseline (*Figure 4—figure supplement 3A*), CCR5 Tg mice show decreased phospho-CREB levels at 3 hr after fear conditioning training (*Figure 4—figure supplement 3C*). Because of the critical role of CREB in memory consolidation (*Bourtchuladze et al., 1994*; *Guzowski and McGaugh, 1997*), these results suggest that the memory deficits caused by CCR5 overexpression in excitatory neurons are due to deficits in memory consolidation processes mediated by CREB (*Silva et al., 1998*).

## *Ccr5*⁺/⁻ mice display faster experience-dependent plasticity in the barrel cortex

To test the hypothesis that CCR5 may also be a neuronal suppressor of neocortical sensory plasticity, we studied experience-dependent plasticity in the barrel cortex. Trimming all of whiskers except for D1 produces both an increase in the responses of layer 2/3 (L2/3) cells located in deprived barrel-columns to stimulation of the spared D1 whisker, and an increase in the size of the spared whisker's representation in the barrel cortex (*Fox, 1992*). Experience-dependent plasticity was assayed by measuring the single unit spike responses to the spared D1 whisker of single L2/3 neurons located in the deprived barrels surrounding the D1 barrel (*Figure 5A*). Although 7-days of single whisker experience is insufficient to produce experience-dependent potentiation in WT barrel cortex (*Glazewski and Fox, 1996*), three measures indicated faster potentiation in *Ccr5*⁺/⁻ mice. First, penetrations in barrels surrounding D1 revealed potentiated responses to D1 stimulation in *Ccr5*⁺/⁻ but not WT mice (*Figure 5A*). Second, a comparison of each cell's response to the spared versus deprived principal whisker response (vibrissae dominance) showed plasticity in *Ccr5*⁺/⁻ (*Figure 5C*, deprived versus undeprived $t_{(21)}$ = 3.222 p<0.01, Student's *t*-test) but not in WT mice (*Figure 5B*, deprived versus undeprived, $t_{(11)}$ = 1.663, p>0.05). Third, a plot of D1 response versus principal whisker (PW) response showed points above the identity line (D1>PW) only for *Ccr5*⁺/⁻ but not WT mice (*Figure 5D*). Comparison of the time course of vibrissae dominance also showed a significant increase in WVDI in *Ccr5*⁺/⁻ mice (0.396 ± 0.051), but not in WT mice (0.111 ± 0.011) after 7 days deprivation [*Figure 5E*; Interaction: $F_{(2,41)}$ = 1.398, genotype, $F_{(1,41)}$ = 8.919, days deprived $F_{(2,41)}$ = 10.20, genotype and deprivation period are significant: t = 3.05, p<0.05, Bonferroni post-tests]. These results demonstrate that *Ccr5*⁺/⁻ mice show faster plasticity in the barrel cortex.

## *Ccr5*⁺/⁻ mice display greater probability and magnitude of LTP in barrel cortical neurons

To investigate whether *Ccr5*⁺/⁻ mice also show enhanced synaptic plasticity in the barrel cortex, a spike-timing dependent protocol was used to induce low levels of LTP in L2/3 cells of WT mice. Whole cell recordings were performed in L2/3 and field stimulation was applied to L4 in the same cortical column (*Figure 6A*), and LTP was induced using a 5 ms pre-post interval as described previously (*Kaneko et al., 2010*). In WT mice, L2/3 cells showed significant LTP in just 20% of cases while in *Ccr5*⁺/⁻ and *Ccr5*⁻/⁻ mice the same protocol produced LTP in 70% of cases (*Figure 6B*, $\chi^2$ = 15.63,

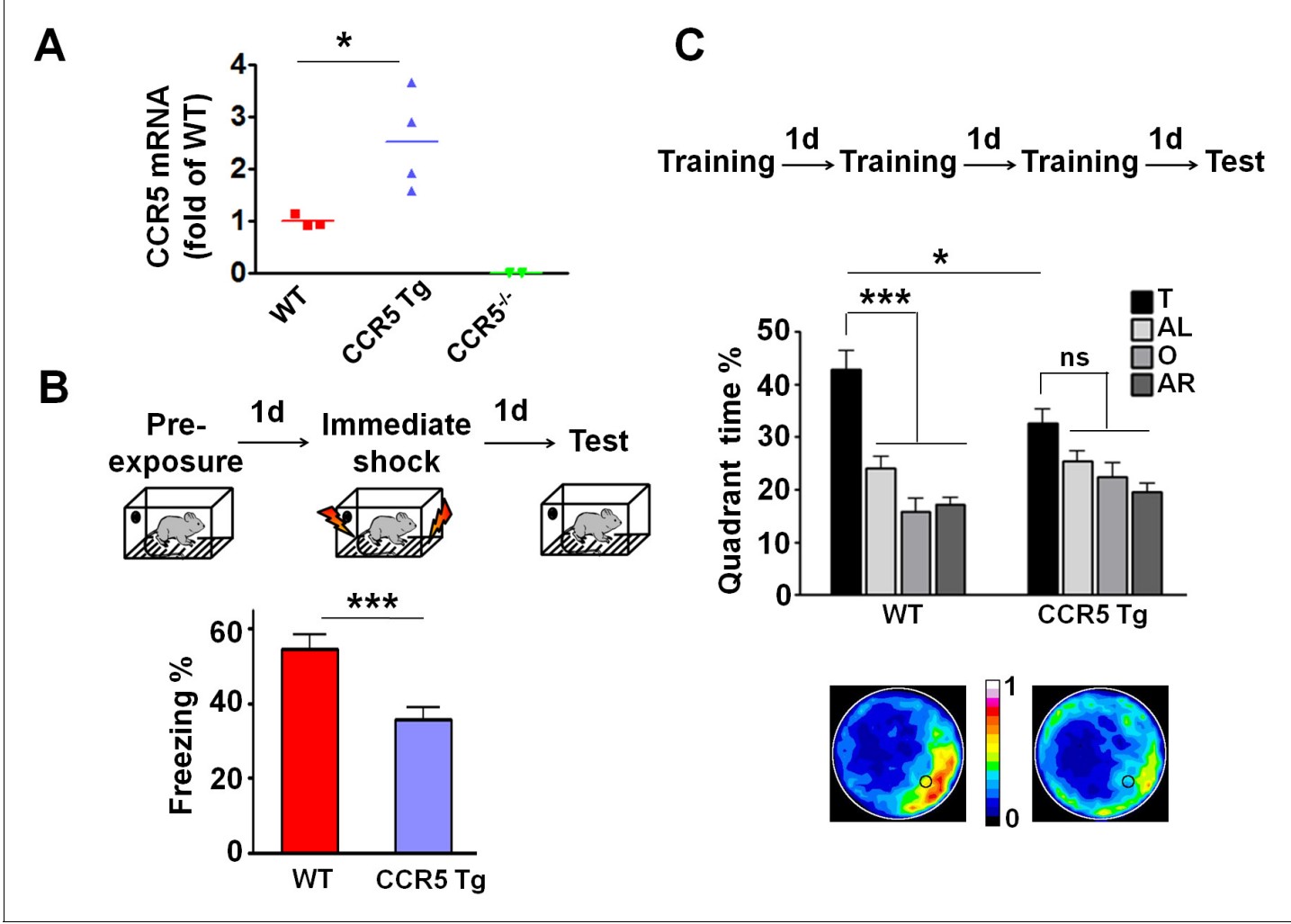

**Figure 4.** CCR5 overexpression leads to learning and memory deficits. (A) Compared to WT mice, CCR5-overexpressing transgenic (Tg) mice showed enhanced *Ccr5* mRNA expression in the hippocampus. (B) In a context pre-exposure fear-conditioning paradigm, CCR5 Tg mice showed contextual memory deficits compared to WT mice (WT n = 13, CCR5 Tg n = 15; ***p<0.001, Student's *t*-test). (C) In the probe test given after 3 days of training in water maze, only WT mice but not CCR5 Tg mice, spent significantly more time in the target quadrant than the other three quadrants; CCR5 Tg mice also had lower searching times in the target quadrant than WT mice, indicating a learning and memory deficit (WT n = 18, CCR5 Tg n = 18; *p<0.05, Student's *t*-test; ***p<0.001, Bonferroni post-tests, Two-way ANOVA with repeated measure). Error bars indicate SEM.

The following figure supplements are available for figure 4:

**Figure supplement 1.** CCR5 overexpression leads to learning and memory deficits.

**Figure supplement 2.** Hippocampal p44/42 MAPK signaling of WT and CCR5 transgenic (Tg) mice.

**Figure supplement 3.** Hippocampal CREB signaling of WT and CCR5 transgenic (Tg) mice.

p<0.001, $\chi$-squared test). On average, the WT mice did not show LTP, while the *Ccr5*[+/-] and *Ccr5*[-/-] mice showed 188.1 ± 1.45% and 164.5 ± 22.5% potentiation, respectively (*Figure 6C*).

### *Ccr5*[+/-] mice display lower release probability, and smaller and less frequent mEPSPs in barrel cortical neurons

Previous studies with HRas[G12V] mice suggest that an increase in ocular dominance plasticity, as well as in learning and memory, is accompanied by a lower synaptic release probability in the cortex

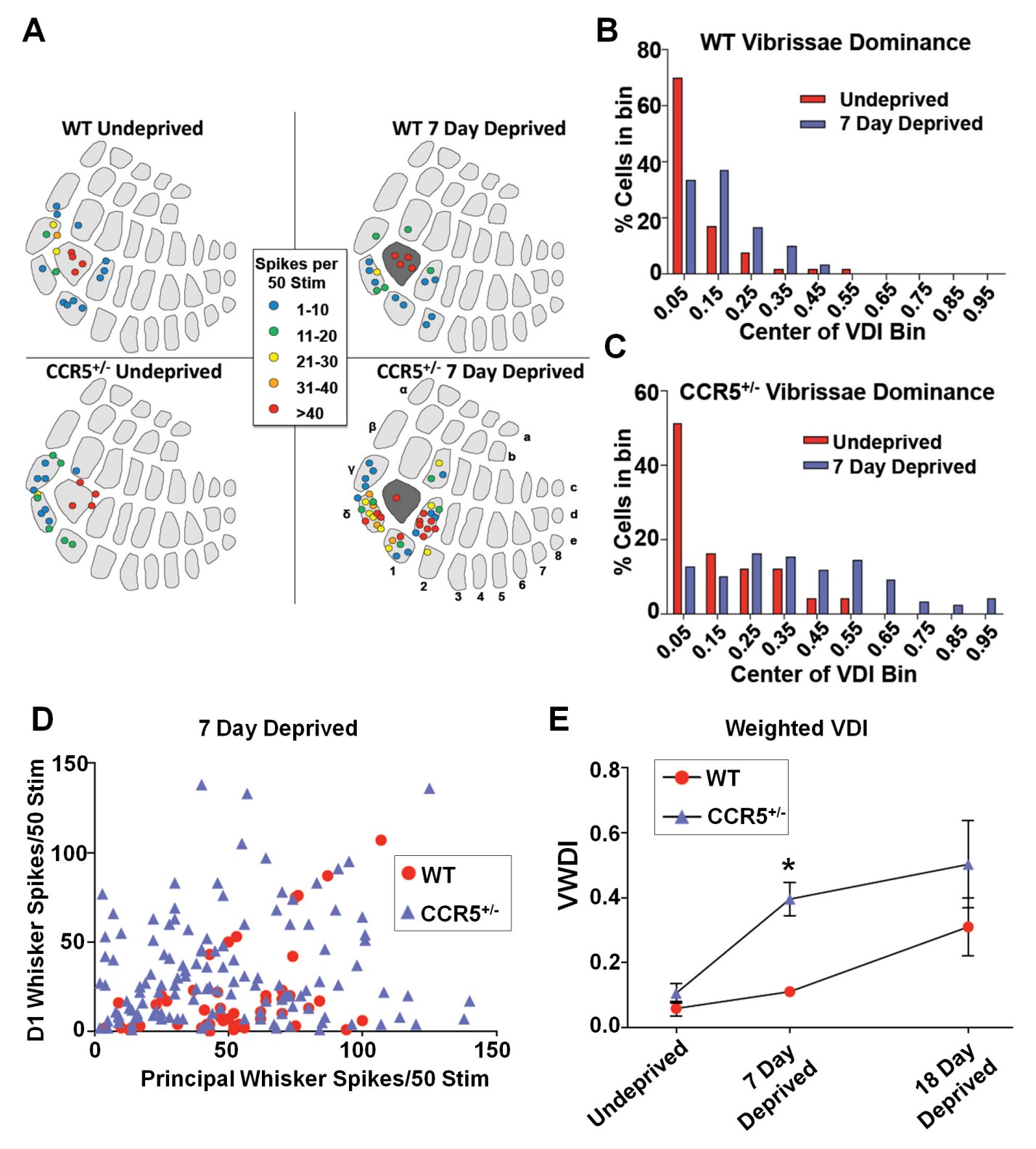

**Figure 5.** *Ccr5*$^{+/-}$ mice display accelerated experience dependent plasticity. (**A**) Recording locations plotted on a standard barrel field map. The barrel shaded dark grey indicates the principal barrel for the spared whisker. The color of each circle represents the average response to D1 whisker stimulation (spikes per 50 stimulations) for cells in L2/3. Deprived *Ccr5*$^{+/-}$ mice showed greater proportion of penetrations responding strongly to D1 stimulation (WT: control vs deprived, p>0.99;*Ccr5*$^{+/-}$: control vs deprived, p=0.0019, binomial test). (**B**) Vibrissae dominance histograms. Vibrissae

*Figure 5 continued on next page*

*Figure 5 continued*

dominance is calculated as D1/(PW + D1) and sorted into 10 bins (see Materials and methods). In WT mice, there is very little shift in the dominance histogram after 7 days of deprivation (n = 14). (**C**) The vibrissae dominance histogram shows a substantial shift right toward D1 dominance in the D1-spared *Ccr5⁺/⁻* mice (blue) compared with undeprived *Ccr5⁺/⁻* mice (red) (n = 24; p<0.01, Student's *t*-test). (**D**) The value of the response to D1 stimulation is plotted against the same L2/3 cell's response to principal whisker (PW) stimulation for mice subject to 7 days deprivation. A large number of *Ccr5⁺/⁻* (but not WT) cells lie above the unity line. (**E**) The average weighted vibrissae dominance index (WVDI) is plotted against deprivation period for WT and *Ccr5⁺/⁻* mice. Naïve WT and *Ccr5⁺/⁻* mice do not exhibit differences in their vibrissae dominance, however after 7 days deprivation there is an increase in WVDI in *Ccr5⁺/⁻* mice but not in WT mice (7-day WT vs *Ccr5⁺/⁻* mice: p=0.0030, ANOVA, Bonferroni post-tests). Error bars indicate SEM.

(*Kaneko et al., 2010*; *Kushner et al., 2005*). Therefore, we investigated the release probability in the barrel cortex of WT, *Ccr5⁺/⁻* and *Ccr5⁻/⁻* mice, by measuring the attenuation rate of NMDA receptor mediated evoked EPSPs (*Figure 7A*) in the presence of the activity-dependent NMDA receptor

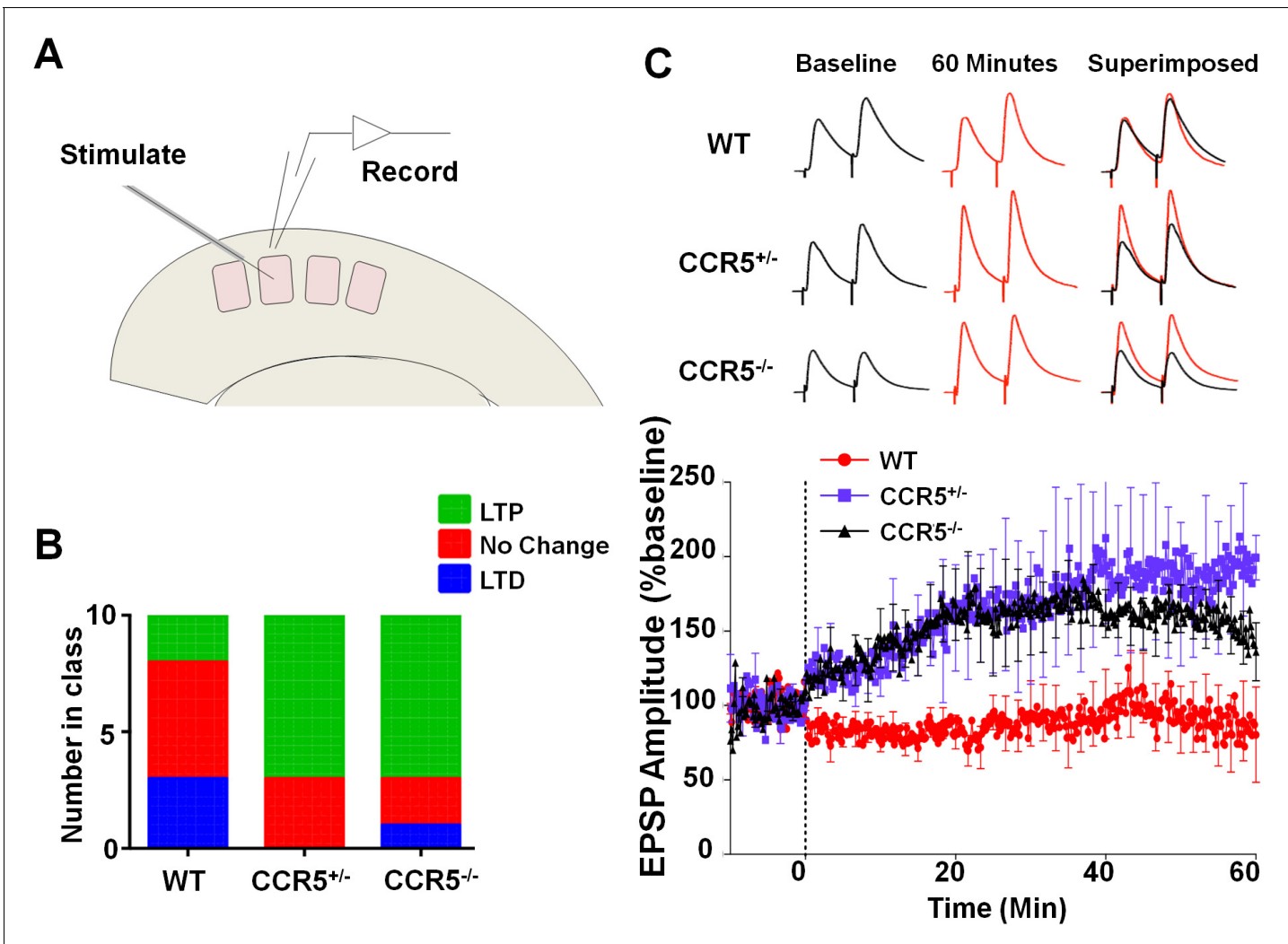

**Figure 6.** *Ccr5* knockout mice exhibit a higher probability of LTP than WT Mice. (**A**) Schematic of LTP in vitro recordings. Whole-cell patch clamp recordings were made from pyramidal cells in L2/3 of the barrel cortex. A stimulating electrode was placed in the center of the barrel immediately below the recording site and the columnar projection from L4 to L2/3 was stimulated. (**B**) When LTP was induced with a spike-timing dependent protocol with a low probability of potentiation in WT mice, *Ccr5⁺/⁻* and *Ccr5⁻/⁻* mice showed a higher incidence of LTP (70% LTP in *Ccr5⁺/⁻* and *Ccr5⁻/⁻* mice vs 20% in WT, p<0.001, χ-squared test). (**C**) Mean amplitude of LTP. Failures and successes are averaged together to give an overall average of all recordings. *Ccr5⁺/⁻* and *Ccr5⁻/⁻* cells had larger LTP than WT cells which on average did not potentiate (n = 10 per group).

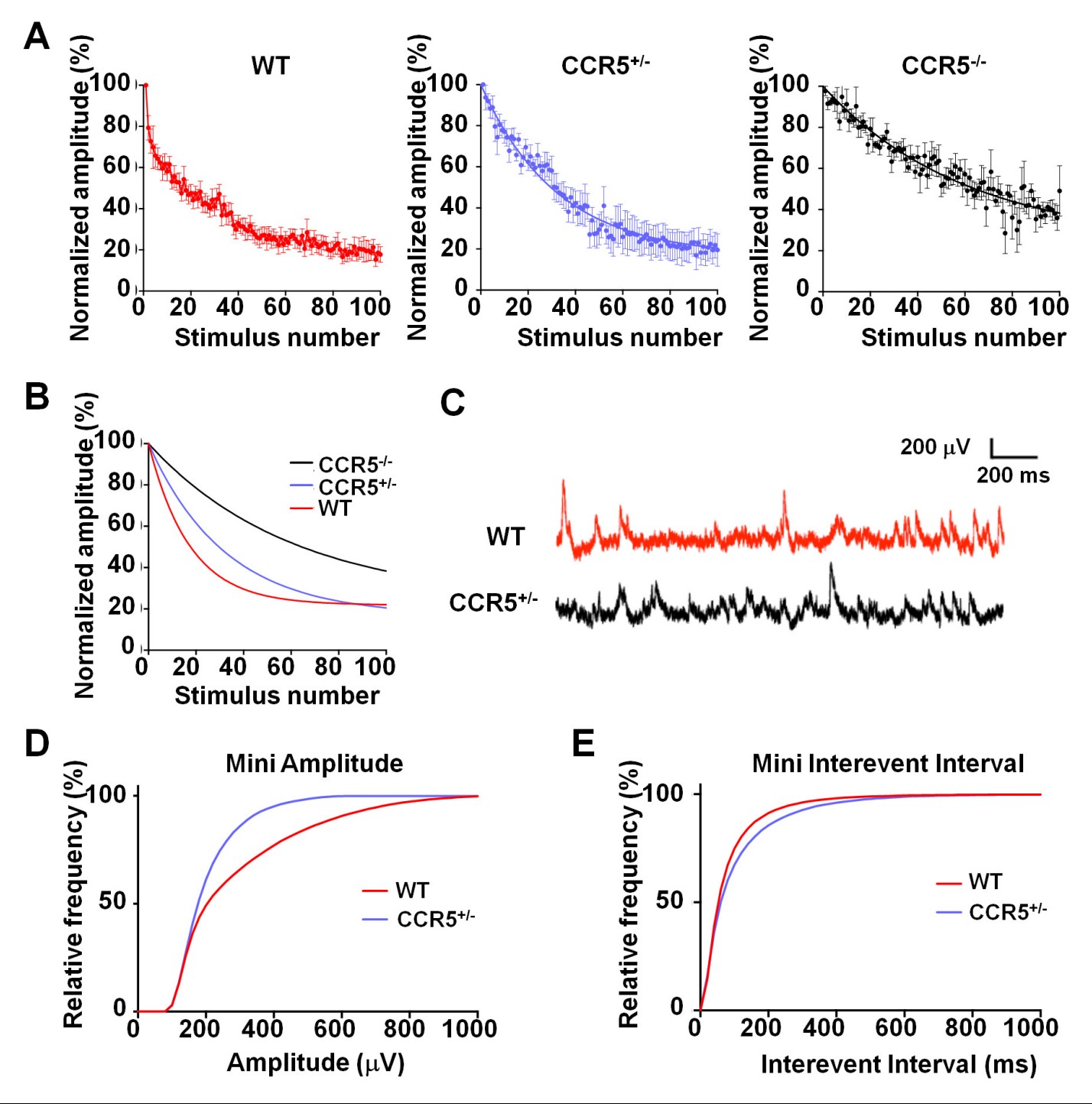

**Figure 7.** *Ccr5* knockout mice exhibit lower release probability, and smaller, less frequent mEPSPs than WT mice. (A) Cells from *Ccr5+/-* and *Ccr5-/-* mice exhibited lower release probability than cells from WT mice. Traces represent normalized amplitude of NMDA-receptor mediated evoked EPSPs in the presence of the use-dependent antagonist MK-801. A faster decrease in the peak EPSP amplitude is indicative of a higher release probability. (B) Single exponential curves were fitted for the data presented in panel **A**, and *Ccr5+/-* and *Ccr5-/-* mice demonstrated lower $P_r$ than WT mice (n = 10 per group; p<0.01, Kruskal-Wallis test). (C) Example miniature EPSPs recordings from WT and *Ccr5+/-* mice. (D) Cells from *Ccr5+/-* mice displayed smaller mEPSPs than WT (WT n = 10, *Ccr5+/-* n = 10; p<0.001, K-S test). (E) Cells from *Ccr5+/-* mice displayed less frequent mEPSPs than WT and therefore greater inter-event intervals (WT n = 10, *Ccr5+/-* n = 10;p<0.001, K-S test).

antagonist MK-801 (*Hessler et al., 1993*). The release probability was significantly lower in *Ccr5⁺/⁻* and *Ccr5⁻/⁻* mice (*Figure 7B*, p<0.01, Kruskal-Wallis test). Consistent with this finding, mEPSPs (*Figure 7C*) in L2/3 cells of *Ccr5⁺/⁻* mice were smaller (*Figure 7D*, $D_{(52)}$ = 0.6154 p<0.001, K-S test) and occurred at a lower frequency compared to WT mice (*Figure 7E*, $D_{(21)}$ = 0.9524 p<0.001, K-S test). Both effects could create greater 'headroom' for LTP and experience-dependent potentiation in *Ccr5* knockout mice.

## Viral knockdown of *Ccr5* expression in adult barrel cortex results in enhanced experience dependent plasticity

To check whether the *Ccr5⁺/⁻* plasticity phenotype in the barrel cortex was dependent on a developmental effect, we also injected shRNA-CCR5 or shRNA-Cont AAV into the barrel cortex of adult C57BL/6N mice. Immuno-histochemistry confirmed that the virus exclusively infected neurons (*Figure 8A* and *Figure 8—figure supplement 1*, 35% infection rate and 100% co-localization of NeuN and GFP). Only shRNA-CCR5 mice, but not shRNA-Cont mice, exhibited a shifted vibrissae dominance histogram after 7 days of single whisker experience (*Figure 8B*, $t_{(10)}$ = 6.485 p<0.001, Student's t-test). Knockdown of *Ccr5* in the adult barrel cortex resulted in an experience-dependent plasticity phenotype that closely mirrored that of the *Ccr5⁺/⁻* mice (*Figure 8C*, *Ccr5⁺/⁻* mice $t_{(20)}$ = 2.939 p<0.01, Student's t-test; viral knockdown: $t_{(10)}$ = 6.485 p<0.001, Student's t-test), demonstrating that CCR5 is a suppressor of neocortical plasticity acting directly on neurons in the adult brain.

## *Ccr5* knockout prevents V3 peptide-induced signaling, plasticity and memory deficits

Our molecular, electrophysiological, sensory plasticity and behavioral results demonstrate that CCR5 is a suppressor for plasticity and learning and memory. Therefore, it is possible that acute activation of CCR5 by HIV coat proteins could contribute to deficits in neuroplasticity and thereby learning and memory. To explore this hypothesis, we tested whether the HIV gp120 V3 loop peptide results in plasticity and learning deficits, and whether CCR5 is responsible for these deficits. The HIV gp120 V3 loop peptide contains the gp120 domain that binds to and activates CCR5 (*Cormier and Dragic, 2002*; *Galanakis et al., 2009*; *Morikis et al., 2007*; *Sirois et al., 2005*). Immunoprecipitation with anti-HA-agarose, to co-precipitate V3-HA peptide and CCR5 in mouse hippocampal lysates, demonstrated that V3 peptide binds to mouse hippocampal CCR5 (*Figure 10—figure supplement 1*).

To examine the effects of V3 peptide on neuroplasticity and the possible role of CCR5 in this, we repeated the cortical LTP study shown in *Figure 6* after pre-incubation for one hour with either 200 pM V3 peptide fragment or saline (control). In cells from WT mice, V3 treatment had a strong effect on the magnitude and probability of LTP (*Figure 9A,B*), completely abolishing any instance of LTP and producing an average depression of 27.8% ($t_{(120)}$ = 26.6, p<0.001, Student's t-test). Cells from *Ccr5⁺/⁻* mice still displayed a similar proportion of LTP (75% V3-treated vs 62.5% control, *Figure 9D*) but showed significantly lower potentiation than in control conditions (127% V3 vs 208% control, *Figure 9C*, $t_{(120)}$ = 42.2, p<0.001, Student's t-test). In contrast, cells from mice lacking CCR5 (*Ccr5⁻/⁻* mice) were not affected by V3 treatment at all, with higher probability of LTP (86% V3 vs 66% control, *Figure 9F*) and a similar magnitude of potentiation (186% V3 vs 182% control, *Figure 9E*). Similar effects of V3 treatment on plasticity were observed in hippocampal slices from WT mice and *Ccr5* knockout mice. Analyses of hippocampal slices, that had been pre-incubated 1 hr with 200 pM V3 peptide, revealed Schaffer collateral fEPSPs LTP deficits between 50 and 60 min post-TBS (theta burst stimulation) in WT (*Figure 9G*, $t_{(14)}$ = 2.53, p<0.05, Student's t-test), but not from *Ccr5⁺/⁻* (*Figure 9H*, $t_{(13)}$ = 1.74, p=0.105, Student's t-test) and *Ccr5⁻/⁻* mice (*Figure 9I*, $t_{(12)}$ = 0.994, p=0.340, Student's t-test). These results demonstrate that V3 peptide causes LTP deficits both in barrel cortex and hippocampus.

To examine the effect of V3 peptide on learning and memory, mice received a single hippocampal V3 peptide infusion (concentration 2 µg/µl, 1 µl peptide to each hemisphere with the infusion speed of 0.1 µl/min) either 30 min before or immediately after contextual fear conditioning (*Figure 10A*). When injected 30 min before fear conditioning, V3 peptide caused contextual memory deficits in C57BL/6N mice (*Figure 10B*, $t_{(10)}$ = 4.04, p<0.01, Student's t-test). In contrast, V3 peptide infusions following training did not affect contextual fear memory (*Figure 10C*, $t_{(18)}$ = 0.92, p=0.37,

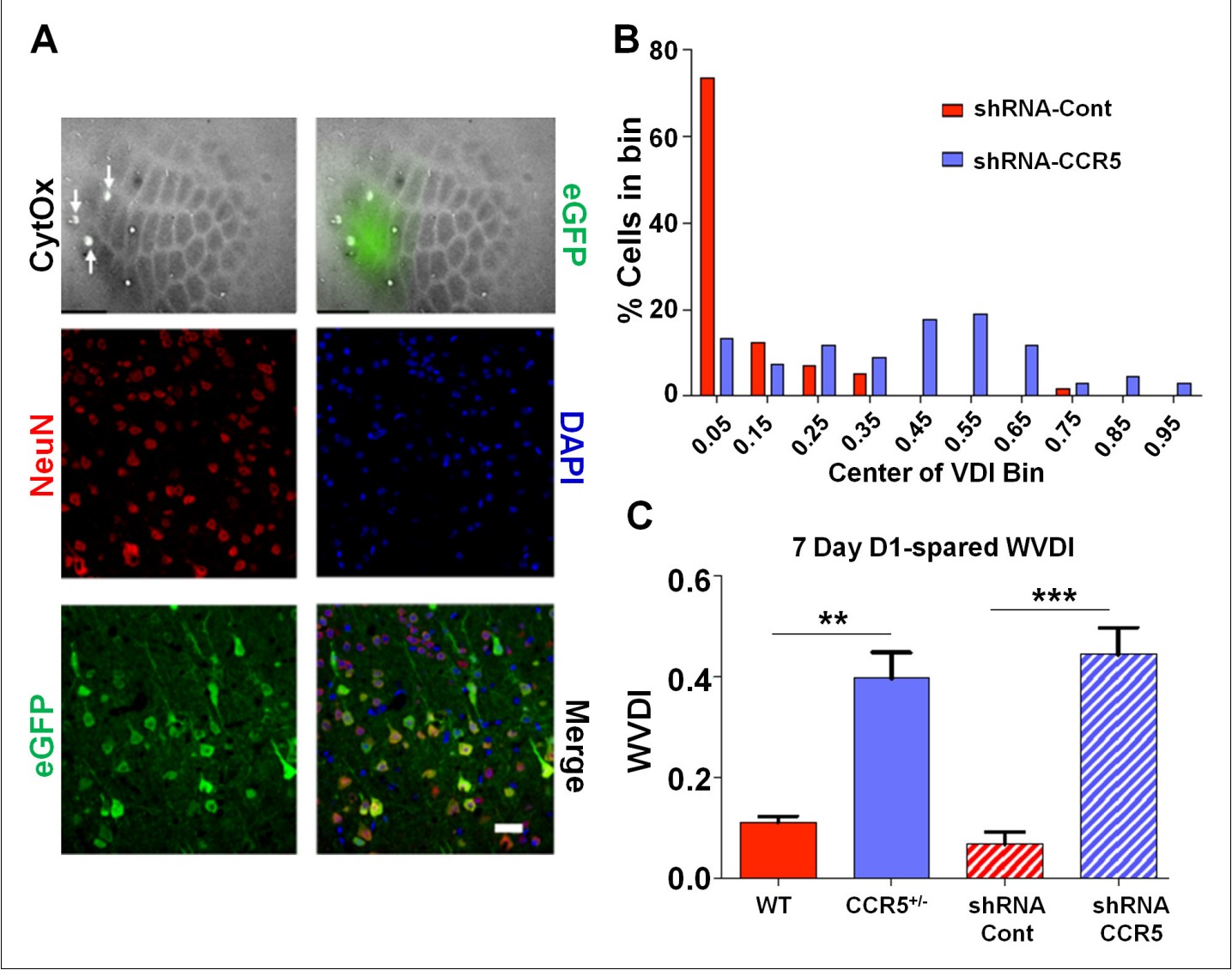

**Figure 8.** Viral knockdown of *Ccr5* in barrel cortex enhances experience-dependent plasticity. (**A**) Specificity and spread of viral infection. Interleaved slices processed for cytochrome oxidase (CytOx) confirmed viral location (GFP) over the recorded area of the barrel cortex. Arrows show lesions marking recording penetrations. Confocal images of the neuronal marker (NeuN) and viral expression (GFP) suggest that the shRNA is expressed exclusively in neurons. Scale bar = 20 μm. (**B**) Vibrissae dominance histogram after 7 days D1-spared single-whisker experience. Compared to control group, knockdown of *Ccr5* leads to a shifted histogram (shRNA-cont n = 6; shRNA-CCR5 n = 6; p<0.001, Student's *t*-test). (**C**) Weighted VDI values comparing *Ccr5*⁺/⁻ mice with viral knockdowns. Viral knockdown has a similar effect to that of constitutive mutants after 7 days deprivation. Error bars indicate SEM.

The following figure supplement is available for figure 8:

**Figure supplement 1.** Specificity of viral infection in the barrel cortex.

Student's *t*-test), indicating that the V3 peptide must be present during training to disrupt learning and memory.

When V3 peptide was infused into the hippocampus of WT, *Ccr5*⁺/⁻ or *Ccr5*⁻/⁻ mice 30 min before training (*Figure 10D*), the V3-induced memory deficit was prevented by the *Ccr5*⁺/⁻ or *Ccr5*⁻/⁻ mutation [*Figure 10E*; Two-way ANOVA, overall (genotype × treatment) interaction: $F_{(2,50)} = 1.0$; Main effect of treatment: $F_{(1,50)} = 11.10$, Post hoc linear contrast: WT/Cont versus WT/V3, $t_{(50)} = 3.25$ p<0.01, Bonferroni post-tests]. To examine whether decreasing *Ccr5* expression specifically in

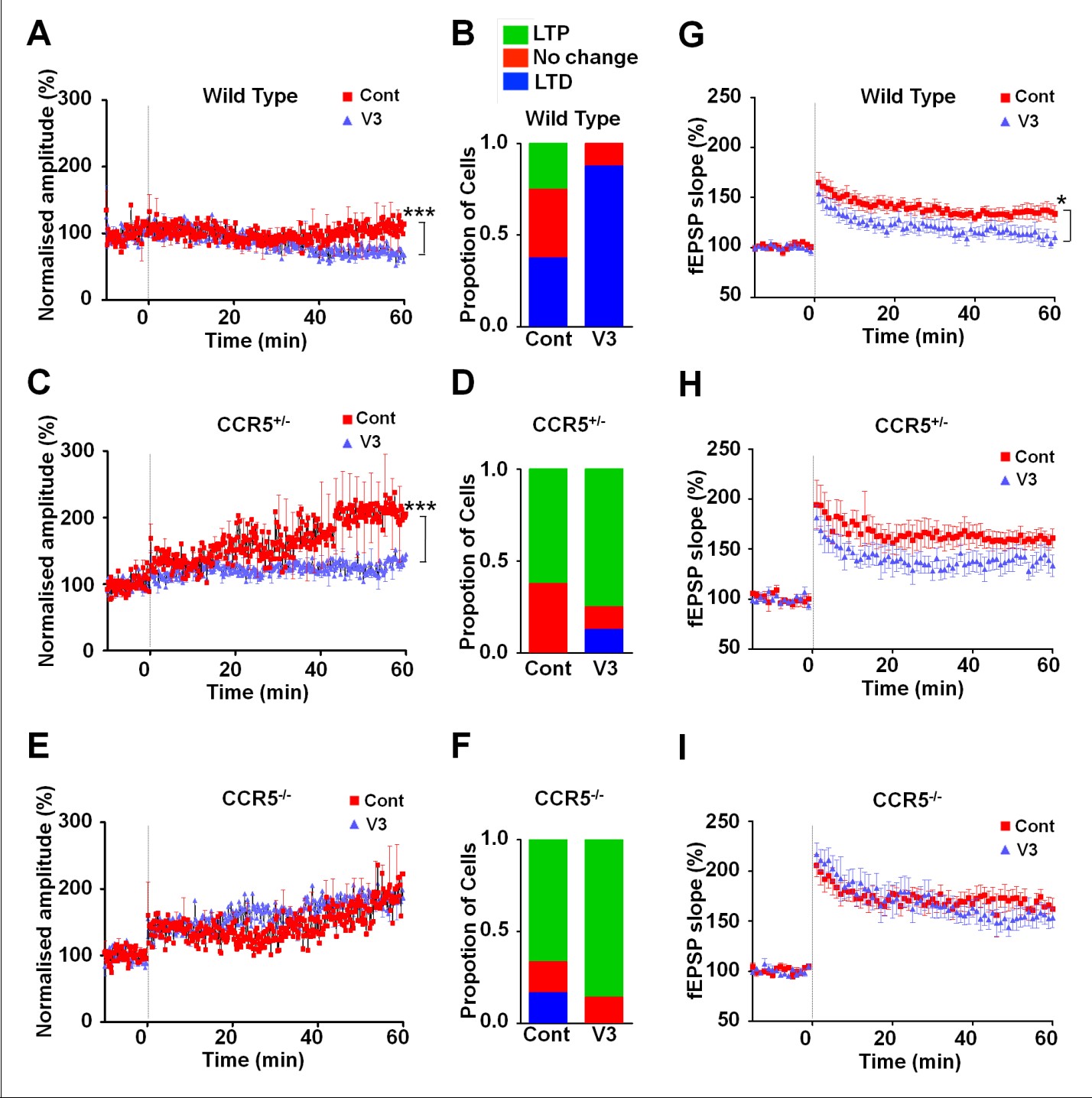

**Figure 9.** *Ccr5* knockout prevents gp120 V3 peptide induced long-term potentiation deficits both in hippocampus and in barrel cortex. (**A**) Whole-cell patch clamp recordings were made from pyramidal cells in L2/3 of the barrel cortex of WT mice. When LTP was induced with a spike-timing dependent protocol, in control conditions the trace showed no significant potentiation, while V3 peptide treatment caused significant depression (n = 8 for both Control and V3; ***p<0.001, Student's *t*-test). (**B**) WT control cells showed equal proportions of LTP, LTD and no change; In contrast, 7 out of 8 V3-treated cells showed LTD. (**C**) Cells from *Ccr5*+/- mice showed strong LTP under control conditions, while V3-treated cells show significantly reduced LTP (n = 8 for both Control and V3; ***p<0.001, Student's *t*-test). (**D**) Although LTP magnitude was reduced in V3-treated cells, the proportion of cells undergoing potentiation in slices from *Ccr5*+/- mice was similar to control. (**E, F**) Both control cells and V3-treated cells from *Ccr5*-/- mice showed strong potentiation, with no significant difference in amplitude and in the probability of LTP (Control n = 6, V3 n = 7; p=0.25). (**G**) Hippocampal CA1 fEPSPs were recorded in hippocampal slices before (baseline) and after 5 TBS (theta bursts stimulation). V3 peptide resulted in LTP deficits in the fEPSP measured during the last 10 min of recordings in hippocampal slices from WT mice (n = 8 for both Control and V3; *p<0.05, Student's *t*-test). (**H, I**) V3

*Figure 9 continued on next page*

*Figure 9 continued*

peptide treatment had no significant effect on $Ccr5^{+/-}$ (control n = 8, V3 n = 7; p=0.105, Student's *t*-test) and $Ccr5^{-/-}$ mice (control n = 7, V3 n = 7, p=0.340, Student's *t*-test). Error bars indicate SEM.

hippocampus is sufficient to block V3-induced memory deficit, shRNA-Cont or shRNA-CCR5 AAV was injected to the hippocampus, and two weeks later, V3 peptide was infused into the hippocampus, and mice were trained 30 min after V3 peptide infusion. Similar to the effects in WT mice, V3 peptide caused memory deficits in shRNA-Cont mice. Importantly, these deficits were prevented by hippocampal *Ccr5* knockdown [*Figure 10F*; Two-way ANOVA, overall (shRNA virus × treatment) interaction: $F_{(1,34)}$ = 0.61; Main effect of treatment: $F_{(1,34)}$ = 7.19; Post hoc linear contrast: shRNA-Cont/Cont versus shRNA-Cont/V3, $t_{(34)}$ = 2.45 p<0.05; shRNA-Cont/Cont versus shRNA-CCR5/Cont, $t_{(34)}$ = 2.55 p<0.05; shRNA-Cont/V3 versus shRNA-CCR5/V3, $t_{(34)}$ = 3.50 p<0.01, Bonferroni post-tests], demonstrating that CCR5 is an in vivo target for the V3 peptide-dependent memory deficits.

Since our results indicate that CCR5 regulates MAPK/CREB signaling, we next determined whether an acute injection of the V3 peptide into the dorsal hippocampal CA1 area results in deficits in MAPK or CREB activation after learning (*Figure 11A*). P44/42 pMAPK and pCREB levels in the dorsal hippocampal CA1 subregion (*Lein et al., 2004*) were measured 1 hr or 3 hr after learning, respectively. Compared to controls, V3 peptide resulted in a significant decrease in pMAPK (*Figure 11B*, $t_{(11)}$ = 2.71 p<0.05, Student's *t*-test), but had no effect on pCREB (*Figure 11C*). A similar decrease in pMAPK was observed when the V3 peptide was infused into the hippocampus of mice injected with shRNA-Cont AAV. Importantly, this decrease in pMAPK activation after learning was ameliorated by hippocampal *Ccr5* knockdown [*Figure 11D*; Two-way ANOVA, overall (shRNA virus × treatment) interaction: $F_{(1,28)}$ = 0.35; Main effect of treatment: $F_{(1,28)}$ = 11.8, Post hoc linear contrast: shRNA-Cont/Cont versus shRNA-Cont/V3, $t_{(28)}$ = 2.85 p<0.05; shRNA-Cont/V3 versus shRNA-CCR5/V3, $t_{(28)}$ = 2.48 p<0.05, Bonferroni post-tests]. Interestingly, unlike the $Ccr5^{+/-}$ mice, that showed enhanced pMAPK activation after learning (*Figure 2B*), compared to shRNA-Cont mice, mice injected with shRNA-CCR5 AAV did not show enhanced pMAPK, probably because in these mice *Ccr5* was knocked down in only the subset of hippocampal neurons transfected by the AAV virus. Altogether, these results indicate that decreasing CCR5 protects against the acute deficits caused by V3 peptide on hippocampal MAPK signaling, synaptic plasticity, and learning and memory, a result consistent with the idea that CCR5 activation contributes to the cognitive deficits triggered by HIV proteins.

## Discussion

CCR5 is a chemokine receptor that plays an important role in inflammatory responses. A learning and memory reverse genetic screen showed that the *Ccr5* knockout results in enhanced learning and memory for contextual fear conditioning. This phenotype was confirmed in other learning and memory tasks including the water maze and social recognition. Importantly, the *Ccr5* knockout did not affect a number of behaviors that could have confounded the learning and memory findings, including anxiety, activity, social interaction, cued conditioning, and shock reactivity. This enhancement in learning and memory is consistent with increases in plasticity at the systems and cellular levels. In the barrel cortex, *Ccr5* knockout results in dramatically accelerated experience-dependent plasticity: $Ccr5^{+/-}$ mice showed robust experience-dependent sensory plasticity at a time point (7-days after whisker removal) when WT mice show no evidence of plasticity. Importantly, temporally and spatially restricted knockdown of *Ccr5*, specifically in adult hippocampus or barrel cortex, also resulted in enhancements in learning and memory as well as experience-dependent plasticity, demonstrating that these effects are not due to changes during development.

MAPK and CREB have been implicated in learning and memory (*Bourtchuladze et al., 1994*; *Roth and Sweatt, 2008*). Our results indicate that the *Ccr5* knockout did not alter MAPK or CREB signaling under baseline conditions. However, *Ccr5* knockout resulted in an increase in MAPK and CREB signaling levels at 1- and 3 hr after training, respectively. Enhanced MAPK or CREB signaling have been associated with enhancements in learning and memory. For example, HRas^G12V mice have enhanced MAPK levels following training, and show increases in LTP and sensory plasticity, as

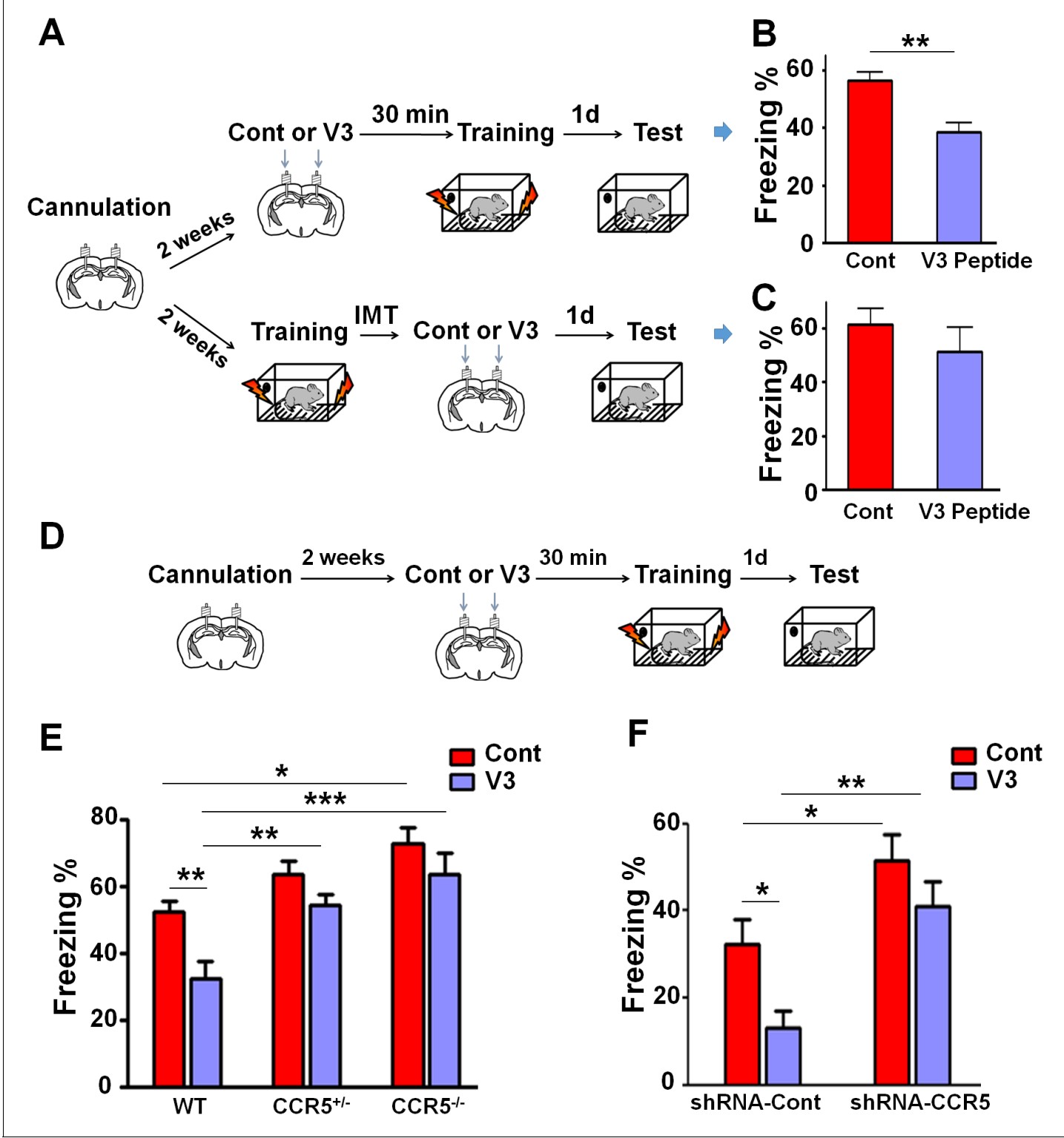

**Figure 10.** Both *Ccr5* knockout and knockdown protect against gp120 V3 peptide induced memory deficits. (A) Schematic of V3 peptide infusion and fear conditioning for the experiments shown in B and C. (B) 30 min after V3 peptide infusion into hippocampus, C57BL/6N mice were trained with fear conditioning. V3 peptide caused contextual memory deficits (Cont n = 6, V3 n = 6; **p<0.01, Student's *t*-test). (C) When V3 peptide was infused into hippocampus immediately after fear conditioning training, no difference was observed between the control group and the V3 peptide group (Cont n = 10, V3 n = 10). (D) Schematic for experiments shown in E and F. (E) 30 min after V3 peptide infusion into hippocampus, mice were trained with fear conditioning. V3 peptide caused contextual memory deficits in WT mice, but *Ccr5* knockout protected against V3-induced memory deficits (WT/Cont

*Figure 10 continued on next page*

*Figure 10 continued*

n = 10, WT/V3 peptide n = 11, *Ccr5*$^{+/-}$/Cont n = 8, *Ccr5*$^{+/-}$/V3 peptide n = 11, *Ccr5*$^{-/-}$/Cont n = 8, *Ccr5*$^{-/-}$/V3 peptide n = 8; *p<0.05, **p<0.01, ***p<0.001, Two-way ANOVA). (**F**) V3 peptide caused contextual fear conditioning memory deficits in mice injected with shRNA-Cont AAV, but hippocampal *Ccr5* knockdown protected against V3-induced memory deficits (shRNA-Cont/Cont n = 10, shRNA-Cont/V3 n = 9, shRNA-CCR5/Cont n = 10, shRNA-CCR5/V3 n = 9; *p<0.05, **p<0.01, Two-way ANOVA). Error bars indicate SEM.

The following figure supplement is available for figure 10:

**Figure supplement 1.** V3-HA peptide binds to hippocampal CCR5.

well as enhancements in learning and memory tested in a number of tasks (*Kaneko et al., 2010*; *Kushner et al., 2005*); Similarly, enhanced CREB signaling is also associated with memory enhancements in multiple tasks and model systems (*Czajkowski et al., 2014*; *Kathirvelu and Colombo, 2013*; *Zhou et al., 2009*) and in barrel cortex plasticity (*Barth et al., 2000*; *Glazewski et al., 1999*).

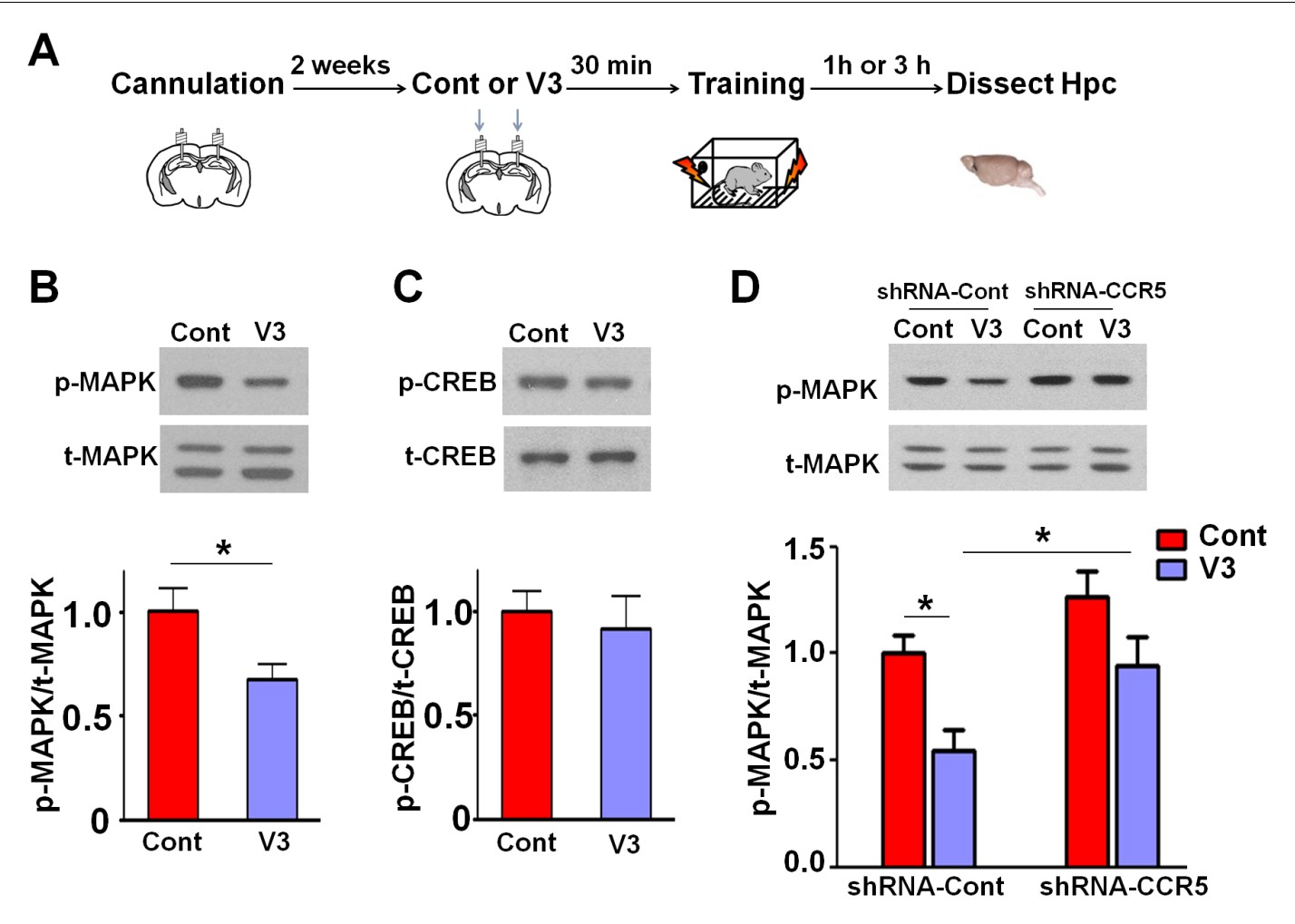

**Figure 11.** *Ccr5* knockdown ameliorates hippocampal p44/42 MAPK signaling deficits caused by V3 peptide treatment. (**A**) The dorsal hippocampal CA1 subregion was extracted 1 hr after training. V3 peptide reduced p44/42 pMAPK levels at 1 hr after fear conditioning (Cont n = 6, V3 n = 7; *p<0.05, Student's *t*-test). (**B**) V3 peptide had no effect on pCREB when dorsal hippocampal CA1 subregion was extracted 3 hr after fear conditioning (Cont n = 5, V3 n = 6). (**C**) V3 peptide reduced p44/42 pMAPK in the dorsal CA1 of mice injected with shRNA-Cont virus at 1 hr after fear conditioning, but *Ccr5* knockdown ameliorated the decrease in p44/42 pMAPK levels observed after learning (n = 8 for each group; *p<0.05, Two-way ANOVA). Error bars indicate SEM.

Altogether, these results indicate that increases in MAPK and CREB signaling are integral to the learning and memory enhancements in the *Ccr5* knockout mice.

*Ccr5* knockout mice show enhanced long-term synaptic plasticity changes both in hippocampus and barrel cortex. For example, our spike pairing protocol in the cortex induced LTP 20% of the time in WT mice and 70% of the time in the *Ccr5* knockout mice. Our neocortical studies also showed that similar to HRas$^{G12V}$ mice (*Kaneko et al., 2010*; *Kushner et al., 2005*), *Ccr5$^{+/-}$* mice also have lower release probability and lower mini-amplitudes. The initial conditions present at the synapses of *Ccr5$^{+/-}$* mice thereby create the headroom for greater pre- and post-synaptic potentiation at the synapse, both of which are important components of plasticity (*Dachtler et al., 2011*; *Hardingham and Fox, 2006*). Together with the increased responsiveness of the MAPK/CREB signaling pathway, the naive state of the synapse under CCR5 hypo-function creates a potent mix for increased synaptic potentiation.

In agreement with the enhanced learning and memory with manipulations that decrease CCR5, transgenic mice that overexpress CCR5 in excitatory neurons show learning and memory deficits, demonstrating that CCR5 acts as a suppressor for plasticity and memory. Although a previous study (*Lee et al., 2009*) reported learning and memory deficits in aged CCR5 mutants, this study used aged mice (12–18 month old) and the control mice with a different genetic background. Our CCR5 transgenic results suggested that CCR5 activation by viral proteins, and subsequent plasticity and memory suppression, may contribute to HIV associated impairments in cognition.

A recent study reported that transgenic mice overexpressing an HIV viral protein (gp120) showed neuronal degeneration and subsequent behavioral deficits that could be ameliorated by a *Ccr5* mutation (*Maung et al., 2014*). Similarly, multiple doses of gp120 in rats also resulted in physiological and cognitive deficits (*Tang et al., 2009*). Interestingly, our study shows that acute treatment (one injection 30 min before training) of HIV gp120 V3 loop peptide, which is known to bind and activate CCR5 in the brain (*Chan et al., 1999*; *Shah et al., 2006*; *Sirois et al., 2005*), was sufficient to induce deficits in a key cellular mechanism for learning and memory (LTP) and also in hippocampus-dependent learning and memory in WT mice. Remarkably, these deficits were prevented by either a viral-mediated *Ccr5*-knockdown or by a *Ccr5* knockout mutation. The V3 loop is important for HIV binding to CCR5 or CXCR4, and LTP deficits caused by the V3 loop peptide had been reported to be blocked by a CXCR4 antagonist (*Dong and Xiong, 2006*). Nevertheless, our results show that both in the barrel cortex and in hippocampus, the LTP deficits caused by the V3 peptide are prevented by the homozygous *Ccr5* mutation, demonstrating that CCR5 plays a critical role in the V3-induced deficits in neuroplasticity. CCR5 antagonists and inhibitors have shown efficacy in reduction of viral load in clinical studies (*Fätkenheuer et al., 2005*; *Gulick et al., 2008*). Our findings support the application of brain permeable CCR5 antagonists, not only as a combination drug in antiretroviral (ARV) therapy, but also as a treatment for cognitive deficits caused by HIV coat proteins.

Previous studies showed that the establishment of long-lasting synaptic changes and long-term memory requires the removal of inhibitory constraints on MAPK/CREB signaling (*Abel et al., 1998*). Our findings with *Ccr5* knockout, region-specific viral knockdown, and neuronal CCR5 over-expression demonstrate that CCR5 functions as a plasticity and memory suppressor by acting on the MAPK/CREB signaling pathway. Our results also suggest that the inappropriate activation of this suppressor by HIV coat proteins contributes to cognitive deficits. Since decreasing CCR5 function leads to robust increases in plasticity and memory, CCR5 provides a novel target for cognitive enhancement, and for the development of treatments for cognitive deficits.

## Materials and methods

### Subjects

3-month old C57BL/6N and *Ccr5* knockout (*Ccr5$^{-/-}$*) mice were purchased from Taconic Farms (Germantown, NY) and tested in a reverse genetic screen (*Figure 1—figure supplement 1*). *Ccr5$^{-/-}$* mice were then bred with C57BL/6N mice to generate *Ccr5$^{+/-}$*. Experimental WT, *Ccr5$^{+/-}$*, and *Ccr5$^{-/-}$* mice (3 to 5 months old) were generated by intercrossing *Ccr5$^{+/-}$* mice. Littermates were used for all experiments (except the initial reverse genetic screen). Experimental *Ccr5*-overexpressing transgenic mice were generated and maintained in the C57BL/6N background. Yfp$^+$/*Ccr5$^{+/-}$* mice were

generated by breeding male Thy1-YFP mice with female *Ccr5*[+/-] mice, and 3 to 6 months old Yfp[+]/*Ccr5*[+/-] mice and their littermates Yfp[+]/*Ccr5*[+/+] (Yfp[+]/WT) mice were used for the spine density experiment. 10-week-old male C57BL/6N mice were purchased from Taconic Farms (Germantown, NY) for shRNA-cont or shRNA-CCR5 AAV injections and for the V3 loop peptide experiments. Mice were group housed with free access to food and water, and maintained on a 12:12 hr light:dark cycle. All experiments were performed during the light phase of the cycle. All studies were approved by the Animal Research Committee at UCLA and University of Cardiff and carried out in compliance with the United Kingdom's Animals (Scientific Procedures) Act 1986 where applicable.

## Fear conditioning

For the reverse genetic screen for remote memory phenotypes, mice were subjected to a training session with a total time of 8 min. The mice were allowed to explore the training chamber for 2 min, and then 3 tone-shock pairs (2 s 0.75 mA shock co-terminated with 30 s tone) were delivered and the 3 tones started at 2, 3 and 4 min. Two weeks after training, the mice were first returned to chambers (context B) which were different from the training chambers (context A) for the cued memory test (tone test), and 90 min later mice were returned to the training chamber for a 5 min test to assess contextual memory. Standard Z scores were used to present the results of the memory screen, which combine both freezing and suppression of activity ratios (SR) scores from the same mice. Suppression of activity ratios (SR) were calculated as SR=(test activity)/(test activity + baseline activity). Z scores for the entire wild type population were calculated as Z score=(individual-population mean)/standard deviation, and the expected value is a mean of 0 and a standard deviation of 1. Z scores for each mutant were calculated as: Z score= average[(%Freezing individual–%Freezing population mean)/(standard deviation of population), (−1)*(SR individual–SR population mean)/(standard deviation of the population)]. So Z scores reflect alterations in both freezing and activity suppression caused by different genetic mutations, with simplified comparisons and improved reliability (*Matynia et al., 2008*).

To test contextual memory for *Ccr5* knockout or knockdown mice, the mice were subjected to a training session with a total time of 5 min. Mice were allowed to explore the training chamber for 2 min, and then 3 shocks (2 s, 0.75 mA) were delivered at 2, 3 and 4 min. To test contextual memory for CCR5 transgenic mice, the mice were either subjected to a daily weak training for 5 days, or to a context pre-exposure fear-conditioning paradigm. During the daily weak conditioning, the mice were allowed to explore the training chamber for 25 s, and then a 1 s, 0.4 mA shock were delivered and mice were removed for the training box 5 s after the shock. The freezing levels during the 25 s period before the shock were used to measure contextual memory. For the context pre-exposure fear conditioning, the mice were pre-exposed to the training context for 7 min on day 1. The mice were brought back to the same context on day 2, and 10 s later received 3 shocks (0.75 mA) with 5 s interval between the 3 shocks. The mice were removed from the training boxes 40 s after the last shock. On day 3, the mice were tested for 5 min in the same training context.

## Morris water maze

In the hidden version of the Morris water maze, mice were trained with two blocks per day for 5 days and each block consisted of two trials with 30 s interval between the trials. In each trial, mice were given 60 s to find the platform. If mice found the platform earlier than 60 s, the trial ended then. If mice failed to find the platform, the trial terminated at 60 s. After each trial, mice were put on the platform for 15 s (*Ccr5*[+/-] and shRNA-CCR5 mice) or 20 s (CCR5 transgenic mice). Probe tests with a time of 60 s were administered after two days of training (*Ccr5*[+/-] and shRNA-CCR5 mice and their controls) or after three days of training (CCR5 transgenic mice and their controls). Testing at day 2 maximized our ability to see enhancements, while testing at day 3 maximized our ability to see deficits. To examine whether mice from different groups were able to learn the Morris water maze task with extended training, a second probe test was performed after five days of training. Mice showing floating behavior during training or probe test were excluded from further training or from data analysis. Floating behavior was judged by mouse swimming speed, and mice with a swimming speed lower than (average speed – 2 × SD (standard deviation)) were excluded.

## Social recognition

Mice were handled for 3 days (2 min handling each day) before social recognition task. During the 3-day social recognition task, mice were first habituated to the testing chamber for 10 min on day 1. On day 2, mice were placed back into the same chamber and habituated for 5 min, and then were allowed to interact for 7 min with an ovariectomized (OVX) female mouse placed under a wired cylinder (training session). On day 3, mice were placed back into the same chamber and habituated for 5 min, and then were allowed to interact with two OVX females (one familiar and one novel) for 5 min (test session). The time mice spent on exploring (sniffing) the OVX mice were hand scored by experimenters blinded of mouse genotypes and of familiar or novel OVX mice.

## Open field

Mice were placed in a novel open field (28 cm × 28 cm × 25 cm), and were allowed to explore it for a period of 20 min. The total distance mice traveled, and percentage time mice spent in the peri- or center region of the open field were analyzed.

## Elevated plus maze

Mice were placed on an elevated plus maze and were allowed to explore it for 5 min. The elevated plus maze has two open arms and two close arms (with walls of 16.5 cm height), and each arm is 29 cm long and 8 cm wide. The percentage of time mice spent in the open arms, and open arm entry times were analyzed.

## *Ccr5* knockdown efficiency measurement

To measure *Ccr5* knockdown efficiency, HEK 293 cells were co-transfected with CCR5-tdTomato plus either shRNA-CCR5 or shRNA-Cont (dsRed) plasmids. HEK293 cells were ordered from ATCC (CRL-1573) and were tested with Mycoplasma Detection Kit (R&D Systems, CUL001B) to make sure there was no contamination. One day after plasmid transfection, cells were imaged to examine the effect of shRNA-CCR5 on CCR5-tdTomato expression. *Ccr5* knockdown efficiency was also tested by measuring *Ccr5* mRNA expression in the hippocampus one month after the injection of AAV containing shRNA-cont or shRNA-CCR5. To measure *Ccr5* mRNA expression, tissue samples were collected from the dorsal hippocampus and total RNA was extracted by RNeasy Mini Kit (Qiagen, Valencia, CA, USA) and treated with DNase (Qiagen). Total RNA was first reverse-transcribed into cDNA using oligo (dT) primers and Superscipt III First-Strand Synthesis System (Invitrogen, Carlsbad, CA, USA), and was then quantified by qPCR. The following primer sequences were used for the *Ccr5* qPCR: 5'GCTGCCTAAACCCTGTCATC 3'(forward) and 5'GTTCTCCTGTGGATCG GGTA3' (reverse). The ribosome protein RPL13A was used as a housekeeping control.

## Immunoblotting for hippocampal samples

Dorsal hippocampus was homogenized with RIPA buffer (Sigma, St. Louis, MO, R0278) supplemented with protease inhibitor cocktail (Sigma, P8340), phosphatase inhibitor cocktail 2 (Sigma, P5726), phosphatase inhibitor cocktail 3 (Sigma, P0044), and 0.5% SDS. After measuring protein concentration with the BCA protein assay kit (Pierce, Rockford, IL, 23225), protein samples were loaded to NuPAGE Novex 4–12% Bis-Tris protein gel (ThermoFisher Scientific, Carlsbad, CA, NP0336BOX), and after separation, proteins were transferred onto polyvinylidene difluoride (PVDF) membranes. The PVDF membranes were blocked with 5% nonfat milk at room temperature for 1 hr and then probed with primary antibodies (phospho-CREB, Cell Signaling 9198, 1:4000 dilution; phospho-p44/42 MAPK, Cell Signaling 9101, 1:10,000 dilution) at 4°C overnight. Membranes were then incubated with secondary antibodies for 1 hr and developed with ECL solutions. The phospho-CREB or phospho- p44/42 MAPK primary antibodies was stripped with Restore western blot stripping buffer (ThermoFisher Scientific, 21059). After stripping, the PVDF membranes were blocked with 5% nonfat milk at room temperature for 1 hr and then probed with primary antibodies (CREB, Cell Signaling 9197, 1:1000 dilution; p44/42 MAPK, Cell Signaling 9102, 1:4000 dilution) at 4°C overnight. Membranes were then incubated with secondary antibodies for 1 hr and developed with ECL plus solutions and scanned with Typhoon 9410 imager and quantified with imageJ.

## Immunostaining for hippocampal samples

Mice were transcardially perfused with 4% PFA (4% paraformaldehyte in 0.1 M phosphate buffer) and after perfusion brains were extracted and incubated with 4% PFA overnight at 4°C. Coronal sections were cut at 50 µm on a microtome and transferred to PBS, then blocked in 5% Normal Goat Serum in 0.1 M PBS and 0.1% TritonX-100 for 1 hr. After blocking sections were incubated in a primary antibody mix (in 0.1 M PBS, 0.2% TritonX-100 and 5% Normal Goat Serum) of rabbit anti-GFP (Abcam AB6556, 1:500 dilution), mouse anti-Neun (Chemokon, MAB377, 1:1000), and rabbit anti-GFAP (Dako Z0334, 1:500) or rabbit anti-Iba1 (Wako 019–19741, 1:500) for two days at 4°C. After 3 × 15 min washes in 0.1 M PBS and 0.1% TritonX-100 the secondary antibodies were applied (in 0.1 M PBS, 0.2% TritonX-100 and 3% Normal Goat Serum): Alexa488 goat anti-rabbit (Invitrogen A-11034, 1:500 dilution), Alexa568 goat anti-rabbit (Invitrogen A-11011, 1:500 dilution), and Alexa647 goat anti-mouse (Invitrogen A-21235, 1:500 dilution). Slices were incubated in the secondary mix for 2 hr at room temperature. After 2 × 15 min washes in 0.1 M PBS and 0.1% TritonX-100, slices were incubated with 4',6-diaminodino-2-phenylindole (DAPI, Life Technologies D-21490, 1:2000) for 15 min, and then were further washed with 0.1 M PBS and 0.1% TritonX-100 for 15 min before mounted onto slides with ProLong Gold antifade mounting media (Life Technologies, P36934). All immunostaining images were acquired with a Nikon A1 Laser Scanning Confocal Microscope (LSCM).

## Immunostaining for cortical samples

Candidate mice were transcardially perfused, as described in the in vivo electrophysiology section, three weeks after injection of AAV. Fixed brains were removed and stored whole in 20% sucrose PBS after 24 hr postfix in 4% formaldehyde/20% sucrose PBS. Coronal sections were cut at 40 µm on a freezing microtome and transferred to PBS, then blocked in 5% Normal Goat Serum in 0.1 M PBS and 0.1% TritonX-100 for 1 hr. After blocking sections were incubated in a primary antibody mix (in 0.1 M PBS, 0.1% TritonX-100 and 3% Normal Goat Serum) of chicken polyclonal anti-GFP (Abcam AB13970, 1:500 dilution) and mouse monoclonal anti-NeuN (Millipore MAB377, 1:100 dilution) for 2 hr at room temperature, 18 hr overnight at 4°C and a further 2 hr at room temperature. After 3 × 30 min washes in 0.1 M PBS and 0.1% TritonX-100 the secondary antibodies were applied (in 0.1 M PBS, 0.1% TritonX-100 and 3% Normal Goat Serum): Alexa488 goat anti-chicken (Life Technologies A11039, 1:200 dilution) and Alexa594 goat anti-mouse (Life Technologies A11032, 1:200 dilution). Slices were incubated in the secondary mix for 3.5 hr at room temperature, and then after a further 3 × 20 min washes in 0.1 M PBS and 0.1% TritonX-100 were mounted in Vectashield DAPI hardset (Vector H1500). The same protocol was applied to tangential slices alternating with slices stained for cytochrome oxidase to localise viral spread across barrels. Coronal sections were visualized using a Leica TCS SP2 confocal microscope and tangential sections were visualized with an Olympus BX61 microscope in both epifluorescent and transmissive brightfield modes. Colocalisation of staining in confocal images was automatically quantified with Imaris F1 7.7.2 (Bitplane, Zurich, Switzerland).

## Hippocampal viral injection surgery

To knockdown *Ccr5* in pyramidal fields of the hippocampus, high titers of Adeno-associated virus (AAV) engineered to knock-down *Ccr5* with an shRNA approach (shRNA-Cont or shRNA-CCR5 viruses, 0.7 µl, $1 \times 10^{13}$ unit/ml) were stereotaxically injected into the hippocampal CA1 sub region of 3 months old C57Bl/6Tac mice through a 30-gauge Hamilton microsyringe at four sites at the following coordinates relative to bregma (mm): AP: −1.8, ML: ±0.8, DV: −1.6; or AP: −2.5, ML: ±2, DV: −1.6. After infusion, the microsyringe was left in place for an additional 5 min to ensure full virus diffusion. After surgery, mice were treated with antibiotics and their health was monitored every day for two weeks.

## Hippocampal V3 peptide infusion

To infuse V3 loop peptide (CTRPNYNKRKRIHIGPGRAFYTTKNIIGTIRQAHC, Disulfide Bridge: 1–35) into hippocampus, two cannula were implanted at the following coordinates relative to bregma (mm): AP: −2.1, ML: ±1.7, DV: −1.6. Two weeks after cannulation, either at 30 min before fear conditioning training or immediately after training, mice were anesthetized and saline (control) or V3 peptide (concentration 2 µg/µl) was infused into hippocampus with the infusion speed 0.1 µl/min and 1

µl solution to each hemisphere. After infusion, the injector was left in place for an additional 5 min to ensure full diffusion.

## V3-HA peptide immunoprecipitation

V3-HA peptide (CTRPNYNKRKRIHIGPGRAFYTTKNIIGTIRQAHCGYPYDVPDYA, Disulfide Bridge: 1–35, from GenScript) and hippocampal CCR5 binding was detected with Pierce co-Immunoprecipitation (Co-IP) kit. Mouse hippocampal tissue was prepared with IP Lysis/Wash Buffer, and was mixed with anti-HA-agarose (Sigma) and incubated overnight at 4°C with or without V3-HA peptide. After elution, the samples were detected with CCR5 antibody (Santa Cruz Biotechnology) with western blot.

## Cortical viral injection surgery

To knockdown *Ccr5* in the barrel cortex, subjects (C57Bl/6N mice; Taconic Farms, Ry, Denmark) aged 2–3 months at time of surgery were anaesthetized with ketamine/xylazine (80/6 mg/kg) and immobilized in a stereotaxic frame (David Kopf Instruments, Tujunga, CA) with a thermostatically controlled heating blanket (Harvard Instruments). The left parietal cranium was exposed and kept moist with sterile cortex buffer. A dental drill was used to make a small craniotomy over the likely location of the D1/D2 barrel (from bregma: AP: −1.5, ML: ±3), the dura resected with a 30 G hypodermic needle and a pulled sharp bevelled glass pipette attached to a Hamilton syringe (Esslab, Essex, UK) was carefully inserted to a depth of 300 µm (DV: −0.3). After 2–3 min 100 nl of high-titre ($1 \times 10^{13}$ unit/ml) AAV was injected over a 5 min period. Fast green was added to the virus to allow visual confirmation of the injection. The incision was closed with sutures and mice were allowed to recover for three weeks before starting the deprivation protocol.

## In vivo electrophysiology

### Subjects

Mice aged between P50 and P90 were anesthetized with isoflurane and maintained by urethane (1.5 g/kg body weight i/p) plus a trace amount of acepromazine. Supplemental doses of urethane (10% of the initial dose) were administered as required to maintain anaesthesia depth. Analgesic (lidocaine) was applied to the ears and scalp. Deprived vibrissae were replaced by attaching the corresponding contralateral whisker to the trimmed stub with cyanoacrylate adhesive. Naïve mice had their whiskers acutely cut and re-attached to minimize mechanical differences between cohorts.

### Surgery

Mice were immobilized in a stereotaxic frame (Narashige, Japan) and body temperature maintained at 37°C with a thermostatically controlled heating blanket (Harvard Apparatus, Kent, UK). A 2 × 2 mm section of the left parietal cranium was thinned with a dental drill over the barrel cortex (0–2 mm caudal from bregma and 2–4 mm lateral from midline). Before each penetration a small fleck of bone was removed from the recording site with a 30 G hypodermic needle just large enough to introduce the electrode.

### Recordings

Recordings were made from barrels corresponding to the spared whisker and its immediate surrounding deprived barrels using carbon fibre microelectrodes. Action potentials were isolated with a window discriminator to provide single-unit recordings, recorded with a Neurolog system (Digitimer, Welwyn Garden City, UK) and digitised with a CED 1401 and Spike2 software (CED, Cambridge, UK) running on a Windows PC.

Stimulation of whiskers was performed with a 3 × 3 matrix piezo-electric stimulator (Cardiff University Mechanical Engineering Centre) driven by a CED 3901 piezo amplifier (CED, Cambridge, UK). Stimuli were applied as a trapezoidal ramp with 10 ms rise time, 10 ms plateau and 10 ms return time, peak deflection was 300 µm (20° whisker deflection). Receptive fields were mapped using pseudorandom sequences in blocks of 10 (9 whiskers and one blank field) at 5 Hz (*Jacob et al., 2010, 2012*).

## Histology

Recording sites were confirmed by histology. After each penetration a small lesion (1 µA DC for 10 s, tip negative) was made at 350 µm estimated depth. At the end of the experiment the mouse was deeply anaesthetized and transcardially perfused with 0.1 M phosphate-buffered saline, followed by 4% formaldehyde in PBS. The brain was carefully removed after fixation and the cortex flattened between two glass slides as previously described (*Strominger and Woolsey, 1987*). After 24 hr postfix in 4% formaldehyde and 20% sucrose in PBS flattened cortices were transferred to 20% sucrose in PBS until sectioning. Tangential sections (35 µm) were cut on a freezing microtome and reacted with diaminobenzidine and cytochrome C to stain for cytochrome oxidase activity (*Wong-Riley, 1979*). This procedure allows for clear identification of the barrels and confirmation of the recording site.

## In vitro electrophysiology

### Slicing procedure

Mice aged between P50 and P60 were killed by cervical dislocation, decapitated and the brain quickly removed and cooled in ice-cold dissection buffer (in mM: 108 choline-Cl, 3 KCl, 26 $NaHCO_3$, 1.25 $NaH_2, PO_4$, 25 D-glucose, 3 Na-pyruvate, 1 $CaCl_2$, 6 $MgSO_4$, 285 mOsm) bubbled with 95% $O_2$/5% $CO_2$. Coronal slices (350 µm thick) were cut on a vibrating microtome (Microm HM650V, Thermo Fisher, Cheshire, UK) and transferred to a holding chamber containing normal artificial CSF (in mM: 119 NaCl, 3.5 KCl, 1 $NaH_2PO_4$, 10 D-glucose, 2 $CaCl_2$, 1 $MgSO_4$, 300 mOsm) bubbled with 95% $O_2$/5% $CO_2$. Slices were incubated at 32°C for 45 min after slicing, then returned to room temperature until recording.

### Recordings

Whole cell recordings were performed at 35–37°C. Barrels were identified in slices under bright field illumination using an Olympus BX50WI microscope. Pyramidal neurons were identified using DIC optics. Recording pipettes (4–10 MΩ) were pulled from borosilicate glass (Clark GC150-F10, Harvard Apparatus, UK) and filled with a potassium gluconate-based recording solution (in mM: 110 K-gluconate, 10 KCl, 2 $MgCl_2$, 2 $Na_2ATP$, 0.03 $Na_2GTP$, 10 HEPES, pH 7.3, 270 mOsm). LII/III pyramidal cells were selected for their characteristic regular spiking behavior under depolarizing current. Recordings were aborted if Vm deviated spontaneously by more than 5 mV, or access resistance deviated by more than 20% during the recording. An Axon Multiclamp 700 B (Molecular Devices, Sunnyvale, CA) in current clamp mode was used as the patch amplifier and signals were telegraphed to and digitised by a CED 1401 with Signal software (CED, Cambridge, UK) running on a Windows PC.

### LTP experiments

A tungsten monopolar stimulating electrode was placed centrally to a selected barrel in layer IV and suitable pyramidal neurons were identified in the area vertically above the stimulating electrode. Once a whole cell recording was established extracellular stimulus was applied at 0.1 Hz, consisting of a pairs of pulses at 20 Hz. The stimulus was of 0.2 ms duration and 1–35 V intensity, designed to produce a monosynaptic EPSP of 3–6 mV in the postsynaptic cell. After 10 min of baseline recording, LTP was induced by pairing a suprathreshold 2.5 ms somatic depolarizing pulse with a single presynaptic stimulus (5 ms pre-post interval). Four runs of 50 paired stimuli at 2 Hz were delivered at 0.025 Hz. After LTP induction the stimulus paradigm was switched back the same as in the baseline recording and one hour of post-paring data were acquired. EPSP amplitudes were calculated as the peak response above the preceding baseline voltage. Significant potentiation was calculated by comparing mean EPSP amplitude at 50–60 mins after LTP induction with mean baseline EPSP amplitude using Student's t-test.

### Release probability experiments

The position of the recording and stimulating electrodes were identical to those for the LTP experiments. Once the recordings were established the perfusion solution was switched to magnesium-free ACSF and an NMDA-receptor mediated response was isolated by the addition of 20 µM CNQX to the bath solution. Single extracellular stimuli were delivered at 0.1 Hz. After a stable baseline had been established, stimulation was halted and 10 µM MK-801 was washed onto the slice for 10 min.

Stimulation was then resumed as before and at least 100 further stimulus trials were recorded, still in the presence of MK-801 and CNQX in Mg-free ACSF. Single and double-exponential fits of the response amplitude over time were performed with GraphPad Prism 5 and 6 (GraphPad, La Jolla, CA). The rate of decrease in the NMDAr-mediated EPSPs is directly related to the release probability of synapses in the observed pathway.

### mEPSP recordings

Recordings were made in layer II/III pyramidal cells. Miniature AMPA receptor-mediated EPSPs were isolated by the bath application of 1 μM tetrodotoxin, 100 μM picrotoxin and 50 μM D-AP5. Action potential blockade was confirmed with the injection of highly-depolarizing square current pulses (0.8–1 nA, 500 ms). 100–500 events were analysed per cell with a template-matching and threshold-crossing method (Axograph X, Berkeley, CA). Well-defined, isolated mEPSPs were identified and used to train a loose-fitting template, with a detection minimum threshold of 2.5 times the RMS baseline noise (*Clements and Bekkers, 1997*). Amplitudes and inter-event intervals were binned and their cumulative distribution compared using a Kolmogorov-Smirnov test.

### Statistical analysis

Results are expressed as mean ± s.e.m. Student's *t*-tests were used for statistical comparisons between groups, unless specified (ANOVA analyses, binomial test, Bonferroni post-tests, χ-squared test, Kruskal-Wallis test or K-S test) in the results or figure legends. $p < 0.05$ indicates significant difference between groups (significance for comparisons: $*p < 0.05$; $**p < 0.01$; $***p < 0.001$). Sample sizes were chosen on the basis of previous studies. No statistical methods were used to predetermine the sample size.

## Acknowledgements

We thank Ayal Lavi, Adam Frank, Michael Sofroniew, Yan Ao, and Kenta Saito for advice and technical support. We thank Antony Landreth, Anna Matynia, Mika Guzman, Cindy Montes, and Aida Amin for their contribution to the reverse genetic screen. This work was supported by grants from the NIMH (P50-MH0779720) to AJS and KDF, and from the MRC (G0901299) to KDF, and the Dr Miriam and Sheldon G Adelson Medical Research Foundation to AJS.

## Additional information

### Funding

| Funder | Grant reference number | Author |
| --- | --- | --- |
| National Institute of Mental Health | P50-MH0779720 | Kevin Fox<br>Alcino J Silva |
| Dr. Miriam and Sheldon G. Adelson Medical Research Foundation | | Alcino J Silva |
| Medical Research Council | G0901299 | Kevin Fox |

The funders had no role in study design, data collection and interpretation, or the decision to submit the work for publication.

### Author contributions

MZ, Conception and design, Acquisition of data, Analysis and interpretation of data, Drafting or revising the article; SG, Conception and design, Acquisition of data, Analysis and interpretation of data, Drafting or revising the article ; SH, YS, SW, YC, MS, Acquisition of data; TKS, Conception and design, Acquisition of data; YN, Contributed unpublished essential data or reagents; DJC, Y-SL, Analysis and interpretation of data; KF, AJS, Conception and design, Analysis and interpretation of data, Drafting or revising the article

## Author ORCIDs

Stuart Greenhill, http://orcid.org/0000-0002-5038-5258

Alcino J Silva, http://orcid.org/0000-0002-1587-4558

## Ethics

Animal experimentation: All experiments were performed during the light phase of the cycle. All studies were approved by the UCLA Institutional Animal Care and Use Committee, also known as the Chancellor's Animal Research Committee (ARC, protocol# 1998-070), and by University of Cardiff and carried out in compliance with the United Kingdom's Animals (Scientific Procedures) Act 1986 where applicable.

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
