## [Decision Letter]

Thank you for submitting your article "CCR5 is a suppressor for cortical plasticity and hippocampal learning and memory" for consideration by *eLife*. Your article has been reviewed by three peer reviewers, and the evaluation has been overseen by a Reviewing Editor and a Senior Editor. The following individual involved in review of your submission has agreed to reveal his identity: Craig Curtis Garner (Reviewer #3).

The reviewers have discussed the reviews with one another and the Reviewing Editor has drafted this decision to help you prepare a revised submission.

The finding that a HIV co-receptor impacts upon learning and memory processes has implications for hippocampal function and may also be relevant to HIV-dementia. The three reviewers provided a number of very positive comments on the study of Ccr5 in hippocampal-dependent plasticity. The findings reported in the study were considered to be novel, interesting and well carried out. The use of overexpression and loss of function of Ccr5 was considered to be appropriate and supportive of the conclusions. It was suggested by the second reviewer that how Ccr5 has effects on excitatory synapses and upon pMAPK and pCREB levels be further considered. These points were brought up, as it is clear how Ccr5 mice with weaker excitatory synapses and lower release probability perform better. We believe you will be able to address these issues in a revised manuscript.

*Reviewer #1:*

This manuscript provides important insight into the role of chemokine receptor CCR5 as a suppressor of learning-related plasticity at the molecular, synaptic, and systems level. Traditionally, CCR5 has been studied as a chemokine receptor that mediates HIV cellular entry, and cognitive deficits in HIV infected individuals have been considered as a byproduct of neuroinflammation. In the present study, Zhou et al. use both loss-of-function (haploinsufficiency of Ccr5 heterozygote mice, shRNA mediated knockdown of Ccr5) and gain-of-function (Ccr5 overexpressing transgenic mice) approaches to show that CCR5 acts as a suppressor of neuroplasticity and memory, and that transgenic overexpression of CCR5 causes deficits in hippocampus-dependent contextual memory and neocortical experience-dependent plasticity. The authors further show that a single injection of HIV coat peptide that forms the ligand for CCR5 is sufficient to cause acute memory deficits that are prevented in Ccr5 null mice.

This is a very impressive and novel study that more than fulfills the criteria for an *eLife* publication. The study addresses the role of CCR5 in learning and memory using multiple approaches that support the overarching conclusion that CCR5 serves as a suppressor of plasticity and memory. I have no issues with the manuscript.

*Reviewer #2 (General assessment and major comments (Required)):*

Understanding the mechanisms that negatively regulate synaptic plasticity and memory processes is an important question. Zhou and colleagues report here that they have identified in a reverse genetic memory screen that hetero- and homozygotic KO mice lacking the GPCR Ccr5 exhibit enhanced memory in multiple memory tasks that involve the hippocampus, as shown here for fear conditioning and the Morris water maze. These behavioral changes are correlated with increased MAPK and CREB signaling following training as well as enhanced hippocampal LTP. In agreement with the hetero- and homozygotic KO data, Ccr5 knockdown in the adult hippocampus results in enhanced memory. Overexpression experiments support that Ccr5 acts in neurons to restrict memory processes. Complementary changes are observed in the barrel cortex, where Ccr5+/- mice or mice in which Ccr5 was knocked-down in neurons show enhanced responses to stimulation of a spared whisker, indicating increased experience-dependent plasticity. Recordings from pyramidal cells in layer 2/3 of the barrel cortex support enhanced LTP in these mice. Other physiological changes include lower release probability and smaller, less frequent mEPSPs in KO mice. Previous findings had shown that HIV coat proteins can bind and activate human CCR5, concomitant with the impaired cognition that can occur in HIV patients. The authors therefore tested acute effects of the gp120 V3 loop peptide on synaptic plasticity and memory processes in mice. They found that loss of Ccr5 or reduction of Ccr5 in hippocampal neurons protects against V3 peptide-induced LTP deficits and memory impairments. This is in agreement with a previous publication by Maung et al. (2014) that reports protection of Ccr5 KO mice against effects of a gp120 peptide. The results extend those findings by determining the timeline of peptide effects and by showing that Ccr5 acts in hippocampal neurons to mediate these effects.

This study reports interesting effects of Ccr5 on synaptic plasticity and memory processes, and presents data that MAPK and CREB signaling are altered upon loss or reduction of Ccr5. The experiments are convincing and allow for a comparison of effects of Ccr5 loss on hippocampal and cortical function. Yet, it remains unclear how Ccr5 functions as a regulator of plasticity and memory processes. Further, it remains open whether the gp120 V3 peptide impacts the same Ccr5-dependent mechanisms that modulate cognition. More detailed comments are provided below.

1) The experiments utilizing viral delivery support that Ccr5 acts in neurons, but the synaptic effects of Ccr5 are unclear. How can the authors explain that mice with weaker excitatory synapses and lower release probability can perform hippocampus-dependent tasks better? The idea of increased "headroom" to potentiate synapses is not compelling as it does not explain why Ccr5 is expressed in the brain if it simply acts to limit excitatory synapse properties. Also, it needs to be tested whether the lower mEPSC frequency in Ccr5 +/- mice is correlated with a reduction in excitatory synapse number. This is a possible alternative or additional cause for this mEPSC phenotype.

2) Through which mechanism can the GPCR Ccr5 impact MAPK and CREB signaling? Also, it is surprising that the activation of only a subset of neurons that occurs in the fear conditioning task is sufficient to substantially increase p-MAPK and p-CREB in total hippocampal homogenates of mice lacking Ccr5. Do the authors assume that the subset of activated Ccr5 +/- neurons undergoes such a massive increase in p-MAPK and p-CREB during learning that this change becomes detectable in total samples?

3) Does the gp120 V3 peptide alter MAPK and CREB signaling in a Ccr5-dependent manner? This will allow assessing to what extent this HIV peptide modulates the same mechanisms through which Ccr5 may restrict plasticity and learning.

4) The authors refer to the publication by Lee et al. (2009) that reports opposite effects of Ccr5 loss on cognitive functions. Yet, they do not discuss this beyond commenting that this is "likely due to genetic background effects". Lee et al. studied these mice in a mixed background while this work analyzes them in a C57BL/6N background. Yet, it is not clear why this difference is a "likely" explanation for this discrepancy. More compelling reasons need to be provided for the opposite phenotype of improved memory processes that is reported here and that is central to this study.

5) The finding that the effect on long-term memory after fear conditioning after shRNA knockdown in Figure 3 is higher than in the heterozygotic Ccr5 KO in Figure 1 and possibly higher than in the full KO in Figure 1—figure supplement 2 raises concerns about off-target effects. The authors can address this by testing whether delivery of Ccr5 shRNA causes effects in KO mice in one of the approaches presented here, e.g. the mEPSC recordings.

6) Does Ccr5 overexpression reduce p-MAPK and p-CREB levels? This would provide evidence that these signaling pathways are indeed linked to Ccr5 effects on neuronal function.

Reviewer #3:

Experiments presented in this study are designed to assess the role of C-C chemokine receptor 5 (CCR5) in learning and memory function. CCR5 is a GPCR involved in the inflammatory response. It is also expressed in CNS. Ligand (RANTES) binding to CCR5 modulates several signaling pathways implicated in learning and memory including adenylyl cyclase, PI3K/AKT and MAPK signaling. CCR5 is also involved in HIV infection. The antagonist maravirco has been found to improve cognition in HIV patents. In this current study, the authors have directly examined the contribution of CCR5 to hippocampal-dependent learning as well as experience-dependent plasticity in the barrel cortex.

Their data nicely shown that decreasing CCR5 function increases not only MAPK/CREB signaling but also LTP and hippocampal-dependent memory in CCR5 +/- and -/- mice. Conversely, over-expression of CCR5 results in memory deficit. In the barrel cortex, decreasing CCR5 enhanced spike timing dependent plasticity and accelerated experience-dependent plasticity. These data support the conclusion that CCR5 normally act to suppress plasticity and memory.

Overall the data presented are of very high quality and strongly support the author's claims. I am very impressed with this provocative study, which provides a strong case for the role of CCR5 as a potent suppressor of learning and memory function. The data presented are presented in a clear and logical manner and the discussion provides a balanced assessment of the database on the literature. The topic is well suited for the readership of *eLife*. I thus recommend publication without revision.

---

## [Author Response]

[…] Reviewer #2 (General assessment and major comments (Required)): […] This study reports interesting effects of Ccr5 on synaptic plasticity and memory processes, and presents data that MAPK and CREB signaling are altered upon loss or reduction of Ccr5. The experiments are convincing and allow for a comparison of effects of Ccr5 loss on hippocampal and cortical function. Yet, it remains unclear how Ccr5 functions as a regulator of plasticity and memory processes.

We respectfully disagree with the reviewer that we did not show “how Ccr5 functions as a regulator of plasticity and memory processes”. In our manuscript, we outline molecular (MAPK/CREB signaling), electrophysiological (LTP and probability of release), structural (spine counts), sensory plasticity and behavioral results that fully account for the function of CCR5 in the brain. Our results provide a large number of convergent and consistent results from multiple CCR5 manipulations (transgenic overexpression, viral knockdown, heterozygous and homozygous germ-line knockouts) that demonstrate the function of this receptor in plasticity and learning and memory. It is important that these results are consistent with a large amount of published data that implicate MAPK/CREB signaling and probability of release studies in plasticity and learning and memory.

Further, it remains open whether the gp120 V3 peptide impacts the same Ccr5-dependent mechanisms that modulate cognition.

The results in the paper address this question with electrophysiological (LTP) and learning and memory experiments, as well as new signaling experiments with *Ccr5* knockdown or germ line knockouts. These studies demonstrated that the same mechanisms modulated by CCR5 (MAPK signaling, LTP and learning and memory) are affected by the gp120 V3 peptide. Together with published evidence, this provides strong evidence that the gp120 V3 peptide affects signaling, synaptic plasticity, and learning and memory mechanism modulated by CCR5.

More detailed comments are provided below.

1) The experiments utilizing viral delivery support that Ccr5 acts in neurons, but the synaptic effects of Ccr5 are unclear. How can the authors explain that mice with weaker excitatory synapses and lower release probability can perform hippocampus-dependent tasks better? The idea of increased "headroom" to potentiate synapses is not compelling as it does not explain why Ccr5 is expressed in the brain if it simply acts to limit excitatory synapse properties. Also, it needs to be tested whether the lower mEPSC frequency in Ccr5 +/- mice is correlated with a reduction in excitatory synapse number. This is a possible alternative or additional cause for this mEPSC phenotype.

The lower mEPSC frequency in *Ccr5* heterozygous (*Ccr5*^+/-^) mice could arise from lower release probability or lower synapse number or both. The independent test of release rate using MK-801 shows that the release rate of *Ccr5*^+/-^ mice is lower (Figure 7). However, we have also addressed the possibility that synapse number could be lower in the *Ccr5*^+/-^ mice by looking at dendritic spines on cortical layer 2/3 neurons (Figure 12). We found that the cortical dendritic spine density of *Ccr5* heterozygotes is very similar to that of their wild-type controls (*Ccr5*^+/-^ mice: spine density = 1.021 spines per micron; wild-type mice: spine density = 1.026 spines per micron). Therefore, lower synapse number is unlikely to account for the lower mEPSC frequency.

Author response image 1.**DOI:**
http://dx.doi.org/10.7554/eLife.20985.027

Consistent with the finding in the cortex, analysis of spine density in the CA1 subregion of hippocampus in *Ccr5*^+/-^ and wild-type mice also show that the dendritic spine density was similar between the wild-type and the *Ccr5*^+/-^ mice (Figure 1—figure supplement 4).

Compared to wild-type mice, the *Ccr5*^+/-^ synapses have a lower release probability on average. However, there is a significant overlap between the distribution of synaptic strengths between mice with less CCR5 and controls. Consequently, many synapses will be just as strong in mice with CCR5 hypo-function as in their wild-type controls. However, the larger “reserve” of weaker release probability synapses in mice with less CCR5 results in a larger set of readily potentiate-able synapses that we believe are responsible for the increased LTP and for the higher learning speed of these mice. The CCR5 ligand RANTES has been shown to modulate glutamate release (Di Prisco et al., 2012; Merega et al., 2015; Musante et al., 2008). A role of RANTES in controlling glutamate release is consistent with our finding that release probability is lower in the *Ccr5*^+/-^ mice. Again, we provide two lines of evidence for this: the MK-801 experiments (Figure 7) and measurements of mEPSC frequency (Figure 7). We also showed that the lower mEPSC frequency is not due to lower synapse number (see Figure 1—figure supplement 4).

We should also point out that there is precedent in the literature for the same association between low release probability, stronger LTP, faster sensory plasticity and enhanced learning and memory. Mice with a HRas^G12V^ mutation show enhanced learning and memory, increased LTP, increased experience dependent sensory plasticity, and importantly, lower release probability (Kaneko et al., 2010; Kushner et al., 2005).

2) Through which mechanism can the GPCR Ccr5 impact MAPK and CREB signaling? Also, it is surprising that the activation of only a subset of neurons that occurs in the fear conditioning task is sufficient to substantially increase p-MAPK and p-CREB in total hippocampal homogenates of mice lacking Ccr5. Do the authors assume that the subset of activated Ccr5 +/- neurons undergoes such a massive increase in p-MAPK and p-CREB during learning that this change becomes detectable in total samples?

CCR5 is a G protein-coupled receptor (GPCR) that is both coupled to the Gαi and Gαq pathway upon activation by endogenous ligands (Blanpain et al., 2002; Mueller and Strange, 2004). Ligand binding to CCR5 modulates p44/42 MAPK signaling (Paruch et al., 2007; Tyner et al., 2005), and p44/42 MAPK can also regulate CCR5 expression (Li et al., 2009). Besides MAPK, CCR5 expression is also modulated by CREB signaling (Kuipers et al., 2008; Wierda and van den Elsen, 2012). Although we did not investigate directly this point in our manuscript, previous results suggest that CCR5 impacts CREB signaling through Gαi pathway and its downstream cAMP and PKA signaling pathways (Delghandi et al., 2005).

"it is surprising that the activation of only a subset of neurons that occurs in the fear conditioning task is sufficient to substantially increase p-MAPK and p-CREB in total hippocampal homogenates of mice lacking Ccr5."

Previous studies have shown that memories are allocated to only a subset of neurons in amygdala, hippocampus, and cortex (Cai et al., 2016; Han et al., 2007; Sano et al., 2014). Nevertheless, previous studies found significant overall increases in p-MAPK or p-CREB after learning (see for example, Atkins et al., 1998; Cammarota et al., 2000; Chen et al., 2012; Kushner et al., 2005). Consistent with previous published findings, we found that activation of a subset of neurons in *Ccr5* knockout mice was sufficient to trigger an overall increase in p44/42 p-MAPK and p-CREB in hippocampal homogenates.

3) Does the gp120 V3 peptide alter MAPK and CREB signaling in a Ccr5-dependent manner? This will allow assessing to what extent this HIV peptide modulates the same mechanisms through which Ccr5 may restrict plasticity and learning.

To test whether V3 peptide modulates the same p44/42 MAPK or CREB mechanisms through which CCR5 suppresses plasticity and memory, we measured hippocampal p44/42 MAPK or CREB signaling after V3 peptide treatment and fear conditioning training (Figure 11). We found that treatment with V3 peptide resulted in reduced phospho- p44/42 MAPK (Figure 11). Importantly, the V3-induced deficits in p44/42 MAPK signaling were ameliorated by *Ccr5* knockdown in the hippocampus (Figure 11). These results demonstrate a signaling mechanism (i.e., reduced phospho- p44/42 MAPK signaling) underlying the protective effect of *Ccr5* knockdown in the V3 peptide induced plasticity and memory deficits.

4) The authors refer to the publication by Lee et al. (2009) that reports opposite effects of Ccr5 loss on cognitive functions. Yet, they do not discuss this beyond commenting that this is "likely due to genetic background effects". Lee et al. studied these mice in a mixed background while this work analyzes them in a C57BL/6N background. Yet, it is not clear why this difference is a "likely" explanation for this discrepancy. More compelling reasons need to be provided for the opposite phenotype of improved memory processes that is reported here and that is central to this study.

There are at least two important factors that could have contributed to the different results observed between the study by Lee et al. and ours: 1) In their behavioral studies, Lee et al. used F2 hybrid mice as controls (B6/129 PF2/J background) for their *Ccr5*^-/-^ (B6J background) mice. There is an extensive literature demonstrating that to determine the effects of a mutation independently from other effects caused by genetic differences between mutants and controls, it is important that the mutant mice and their controls are in the same genetic background. For example, the hybrid mice used as controls in the Lee et al. study out-perform inbred mice, such as the B6J strain (genetic background of the *Ccr5*^-/-^ mutants in the Lee et al. study), in learning tasks. We also found that in a fear conditioning task, B6/129 hybrid mice given very weak training (one 0.5mA shock) showed similar memory levels (Wiltgen and Silva, 2007) as the C57BL/6NTac mice in our study that received very strong training (3x0.75mA shocks) (Figure 1, Figure 3, and Figure 1—figure supplement 2 in our current study). In contrast to the Lee et al. study, we used wild type (WT) littermates as controls for the *Ccr5*^-/-^ and *Ccr5*^+/-^ mutant mice we studied. In our study, both WT controls and *Ccr5* mice are in the same C57BL/6NTac background. 2) Additionally, Lee et al. used aged mice (12-18 month old) in their study, while we used adult mice (3-5 months old) for our experiments. Cell surface expression of CCR5 has been reported to change with age (Abraham et al., 2008), which may change the role of CCR5 in learning and memory during aging. We have modified our comment in the Discussion section as following "Although a previous study (Lee et al., 2009) reported learning and memory deficits in aged CCR5 mutants, this study used aged mice (12-18 month old) and the control mice with a different genetic background".

5) The finding that the effect on long-term memory after fear conditioning after shRNA knockdown in Figure 3 is higher than in the heterozygotic Ccr5 KO in Figure 1 and possibly higher than in the full KO in Figure 1—figure supplement 2 raises concerns about off-target effects. The authors can address this by testing whether delivery of Ccr5 shRNA causes effects in KO mice in one of the approaches presented here, e.g. the mEPSC recordings.

We respectfully disagree with the reviewer. It is not uncommon to observe a certain amount of variability in behavioral performance across experiments (which could be due to factors such as environment, season changes, different experimenters, etc.). Because of behavioral variability, in all of our experiments we always compared the manipulation studied (knockout, knockdown) with appropriate experimental cohort controls.

For example, in the fear conditioning figures (Figure 13), we show here the 8 different fear conditioning experiments we have performed with mice with lower levels of CCR5. Although the levels of mouse freezing vary between experiments, the freezing levels of the shRNA experiment fall within this range, and importantly, *Ccr5^+/-^, Ccr5^-/-^*mice or mice with *Ccr5* knockdown consistently showed enhanced memory when compared to their controls that were trained and tested at the same time. These results provide strong evidence that lowering *Ccr5* levels result in enhanced learning and memory.

Author response image 2.Fear conditioning results of WT and *Ccr5^+/-^, Ccr5^-/-^* mice or mice with *Ccr5* knockdown.(**A**) Mice were tested 1 day after fear conditioning training. (**B**) Mice were tested 1 week after fear conditioning training. (**C**) Mice were tested 2 weeks after fear conditioning training. A dash line marking the freezing at 40% level was shown in the figures to help compare mouse freezing across different experiments.**DOI:**
http://dx.doi.org/10.7554/eLife.20985.028

6) Does Ccr5 overexpression reduce p-MAPK and p-CREB levels? This would provide evidence that these signaling pathways are indeed linked to Ccr5 effects on neuronal function.

Our results show that CCR5 transgenic mice that overexpress CCR5 in the excitatory neurons show decreased phospho-CREB levels 3 hours after fear conditioning training (Figure 4—figure supplement 3; we found the opposite result in *Ccr5* knockouts). Because of the critical role of CREB in memory consolidation (Bourtchuladze et al., 1994; Guzowski and McGaugh, 1997), this decrease in phospho-CREB accounts for the memory deficits of these transgenic mice.